# Towards Robust Scale-Invariant Mutual Information Estimators

## Abstract

Mutual information (MI) is hard to estimate for high dimensional data, and various estimators have been proposed over the years to tackle this problem. Here, we note that there exists another challenging problem, namely that many estimators of MI, which we denote as $I(X;T)$, are sensitive to scale, i.e., $I(X;\alpha T) \neq I(X;T)$ where $\alpha \in \mathbb{R}$. Although some normalization methods have been hinted at in previous works, there is no in-depth study of the problem. In this work, we study new normalization strategies for MI estimators to be scale-invariant, particularly for the Kraskov–Stögbauer–Grassberger (KSG) and the neural network-based MI (MINE) estimators. We provide theoretical and empirical results and show that the original un-normalized estimators are not scale-invariant and highlight the consequences of an estimator's scale-dependence. We propose new global normalization strategies that are tuned to the corresponding estimator and scale invariant. We compare our global normalization strategies to existing local normalization strategies and provide intuitive and empirical arguments to support the use of global normalization. Extensive experiments across multiple distributions and settings are conducted, and we find that our proposed variants KSG-Global-$L_\infty$ and MINE-Global-Corrected are most accurate within their respective approaches. Finally, we perform an information plane analysis of neural networks and observe clearer trends of fitting and compression using the normalized estimators compared to the original un-normalized estimators. Our work highlights the importance of scale awareness and global normalization in the MI estimation problem.

## 1 Introduction

Mutual information (MI), is a fundamental measure of dependency between two variables, which has become pivotal in various machine learning domains, including generalization (Xu & Raginsky, 2017; Bu et al., 2019; Russo & Zou, 2020), representation learning (Bachman et al., 2019; Tschannen et al., 2020) and fairness (Wang et al., 2023; Roh et al., 2020). Estimating MI for high-dimensional continuous variables (Xu et al., 2020) is particularly challenging, due to the hardness of accurately estimating the probability distribution in high dimensions (Goldfeld & Greenewald, 2021). For example, traditional estimators like Kraskov–Stögbauer–Grassberger (KSG) (Kraskov et al., 2004), rely on distance metrics, and for high dimensional data, the distances would have less variation due to the curse of dimensionality.

In this paper, we highlight a critical but underexplored factor that leads to inaccuracies in MI estimation: the scale of the variables ( i.e., $|X|$). Specifically, when considering the mutual information $I(X;\alpha T)$, where $\alpha \in \mathbb{R}^+$ is a scaling factor, we demonstrate that when $\alpha \ll 1$ or $\alpha \gg 1$, the MI estimates can deviate significantly from the true value. This is problematic since by definition, $I(X;\alpha T) = I(X;T)$ for any two continuous random variables (RVs) $X$ and $T$. Moreover, a stronger result states that $I(X;f(Y)) = I(X;T)$ for any continuous and invertible transformation $f$ (Cover & Thomas, 2006). In this paper, we mainly focus on the specific impact of scale.

Most mutual information estimators, including the widely adopted KSG estimator (Kraskov et al., 2004) and its subsequent variants (Gao et al., 2017), lack scale invariance—a limitation that we rigorously demonstrate in this study. We provide a theoretical analysis explaining why this deficiency arises. We also show the

binning estimator (Paninski, 2003) for MI can be scale invariant when the number of bins used is fixed. However, the binning estimator itself is not well-suited for estimating high-dimensional continuous variables. Recently the mutual information neural estimator (MINE) (Belghazi et al., 2018) was proposed, which is a neural network-based estimator of MI that makes use of its Donsker-Varadhan (DV) representation. We demonstrate theoretically that ideally, MINE should be scale-invariant, but MINE fails in practice due to limitations introduced by stochastic gradient descent optimization.

Despite numerous surveys that have explored various methods of MI estimation (Walters-Williams & Li, 2009; McAllester & Stratos, 2020; Paninski, 2003), the critical importance of normalization (preprocessing) has been largely overlooked. A natural solution to ensure scale invariance is to pre-process the data using standard normalization, where each dimension is adjusted to have a variance of 1, and we refer as *local normalization.* This pre-processing step was hinted in (Kraskov et al., 2004) for the KSG estimator. Local normalization also has been commonly applied as a preprocessing step in many deep learning studies involving mutual information perspective (Hjelm et al., 2019; Xie et al., 2024). However, local normalization treats each dimension independently and normalizes them to have a variance of 1, which, as we demonstrate in Section 5.1.1, does not work well in the high-dimension setting especially in neural networks, across two separate experiments. This is because most high-dimensional feature representations in neural networks always contain some noisy dimensions, which are of low energy and contain irrelevant features. Thus, amplifying these low energy dimensions can lead to suboptimal MI estimates. We also note that the recent work by (Czyż et al., 2023), in addition to trying out local normalization approaches, also studied other preprocessing methods including the transformation of the margin distribution to uniform distribution (via converting to rank). We note that this conversion step also brings all individual dimensions to equal importance like local normalization, and thus would have the same pitfalls in this scenario.

To address this issue, in our work, we propose a set of *global normalization* approaches. Unlike local normalization, global normalization preserves the relative energies between the different dimensions, and thus avoids scaling up low-energy noisy dimensions. Our proposed estimator modifications do not only include new normalization approaches, however, and often also have an additional maximization step, which helps bias our estimators better. It is well known that KSG and other MI estimators have a tendency to have negative bias Czyż et al. (2023), especially in high dimensions. Our normalization approaches for KSG incorporate this observation via an additional maximization step, which also follows intuitively from one of our theoretical observations in Proposition 3.

We now summarize our contributions:
- We propose novel scale-invariant extensions of KSG and MINE-based estimators that effectively address the one-sided scale-invariance issue and substantially improve estimator accuracy. To the best of our knowledge, our work is the first comprehensive analysis of the effect of scale and various normalization methodologies, some of which are introduced for the first time in this work.
- We demonstrate that the KSG-Global-$L_\infty$ and MINE-Global-Corrected variants consistently produce the most accurate estimations within their respective approaches, across a broad range of experiments involving synthetic data, which are targeted towards the high-dimensional and low-data regime. These experiments include multiple types of transformations, noise injections, and changes in dimensionality.
- We explore the dynamics of MI between inputs $X$ and hidden layers $T$ during neural network training. Our results highlight that unnormalized estimators significantly confound the scale of $T$ in their estimates, while our normalized approaches can often capture distinct phases of training, such as fitting and compression.

The rest of the paper is organized as follows. In Section 2, we first provide formal definitions of information measures and briefly review the common MI estimators employed in our study. In Section 3, we theoretically study scale-invariance behaviour of three estimators: binning, KSG and MINE. In Section 4, we introduce our proposed scale-invariant variants of KSG and MINE using new normalization strategies. In Section 5.1, we provide additional empirical motivation for each aspect of the proposed variants of KSG and MINE. In section 5.2, we conduct extensive experiments on synthetic datasets and rigorously test our proposed estimators against the non-normalized and standard locally-normalized estimators. In 5.3 we conduct experiments on three real datasets, which mainly includes monitoring the mutual information measures during neural network

training. Comparisons are made against both the original estimators and standard normalization approaches (local normalization). Finally, we summarize our findings and discuss their implications in Section 7.

## 2 Background

### 2.1 Mutual Information

Mutual information of two variables is a statistical measure that quantifies the mutual dependence between two random variables. Specifically, it measures the amount of information obtained about one random variable through the observation of another. To understand mutual information, it is essential to first examine another foundational concept, Shannon entropy. Shannon entropy represents the intrinsic informational uncertainty associate with a probabilistic system. Given a continuous random variable $X$ with a probability density function $f$ from a set $\mathcal{X}$, the continuous entropy $h(X)$ is defined as:

$$h(X) := - \int_{\mathcal{X}} f(x) \log f(x) \, dx \tag{1}$$

Then, the mutual information between continuous random variables $X$ and $Y$ is given by:

$$I(X;Y) = h(X) + h(Y) - h(X,Y) \tag{2}$$

where $h(X,Y)$ represents the joint differential entropy of $X$ and $Y$, defined as $h(X,Y) = - \int_{\mathcal{X},\mathcal{Y}} f(x,y) \log f(x,y) \, dxdy$. Mutual information can be interpreted as the reduction in the uncertainty of $X$ due to the knowledge of $Y$, or equivalently, as the amount of information that $X$ and $Y$ share.

In the case of jointly continuous random variables, the mutual information can be expressed in terms of Kullback–Leibler (KL-) divergence

$$I(X;Y) = D_{\mathrm{KL}} \left( P(X,Y) \| P(X) \otimes P(Y) \right), \tag{3}$$

where $P(X) \otimes P(Y)$ is the dot product of two marginal distributions $P(X)$ and $P(Y)$, $P(X,Y)$ is their joint distribution. $D_{\mathrm{KL}}$ is defined as

$$D_{\mathrm{KL}}(P \| Q) := \mathbb{E}_P \left[ \log \frac{dP}{dQ} \right]. \tag{4}$$

In practice, estimating the true distribution of continuous random variables is challenging, especially for high-dimensional data. In the following section, we will discuss various non-parametric MI estimators, which estimate the distribution of random variables and subsequently compute estimated mutual information.

### 2.2 Mutual Information Estimators

The overall setting of the MI estimation problem is as follows. We are given two RVs $X, T \sim P(X,T)$ and sampled data $S = \{(X_1, T_1), ..., (X_n, T_n)\}$. Our estimators are denoted in the form $\widehat{I}_{est}^n(X;T)$, where $est$ denotes the name of the estimator.

In this section, we present several widely-used nonparametric MI estimators that are studied in our work and have been extensively applied in other research.

**Binning Estimator:** Also called histogram based estimator in many research. This method represents the most elementary technique for estimating mutual information. To estimate MI, the continuous random variable is discretized into bins, counting the number of samples that fall into each bin, and computing the probability density (Paninski, 2003). The binning estimator for $n$ samples can be expressed as: $\widehat{I}_{bin}^n(X;T) = H_{bin}(X) + H_{bin}(T) - H_{bin}(X,T)$. where $H_{bin}(X)$ represents the binned entropy given a RV $X$, such that $H_{bin}(X) = - \sum_i P(X_i) \log P(X_i)$. Let $n(X_i)$ be the number of samples that fall in $i$-th bin of $X$, and $N$ is the total number of data points. Then we have $P(X_i) \approx n(X_i)/N$ for binning method. Similarly, we represent binned joint entropy as $H_{bin}(X,T) = - \sum_{i,j} P(X_i, T_j) \log P(X_i, T_j)$, and $P(X_i, T_j) \approx n(X_i, T_j)/N$.

**Kraskov–Stögbauer–Grassberger (KSG) Estimator:** Another popular non-parametric approach to estimate MI in high dimensions is the KSG estimator in (Kraskov et al., 2004). Unlike the binning estimator, the KSG estimator uses the $k$-nearest neighbor ($K$-NN) statistic to estimate the probability function of continuous random variables, which also uses the joint entropy decomposition method to estimate MI. The KSG estimator effectively uses the k-nearest neighbor distances to estimate the various entropies involved in the joint-entropy decomposition of $I(X;T)$. To define the KSG estimator, we will need some pre-requisites. First, let the $k$-NN distance $\rho_{k,i,p}$ be defined as the distance from $(X_i, T_i)$ to $k$-th nearest neighbor in the joint space $(X,T)$ as measured in $l_p$ distance. We now denote $n_{x,i,p} = \sum_{j \neq i} \mathbb{I}\left\{\|X_j - X_i\|_p \leq \rho_{k,i,p}\right\}$ as the number of neighbors of the $i$-th sample $X_i$ within a specified distance under the $l_p$ norm, and similarly for $T$ as $n_{t,i,p}$. Eventually, the KSG estimator yields the following estimate of $I(X;T)$:

$$\widehat{I}^n_{KSG}(X;T) = \psi(k) + \psi(n) - \frac{1}{k} - \frac{1}{n}\sum_{i=1}^{n}\left(\psi(n_{x,i,\infty}) + \psi(n_{t,i,\infty})\right), \tag{5}$$

where $\psi(x)$ is the digamma function (i.e., $\psi(x) = \Gamma(x)^{-1}d\Gamma(x)dx$).

In (Gao et al., 2017), authors proposed a bias-improved KSG (**BI-KSG**) that performs better than KSG when $N$ is small and $X$ and $T$ are not independent. It is also important to note that many other variants of KSG and other estimators (Pál et al., 2010; Gao et al., 2015) use $k$-NN approach.

**Mutual Information Neural Estimator (MINE):** In our work, we utilize neural network based MI estimators, specifically Mutual Information Neural Estimation (Belghazi et al., 2018). This approach estimates mutual information by using a dual representation of of the KL-divergence, known as Donsker-Varadhan (DV) representation (Donsker & Varadhan, 1983). Given RVs $X \sim P(X)$, $T \sim P(T)$, and $(X_i, T_i) \sim P(X,T)$, we express equation 3 in terms of DV representation as:

$$I(X;T) = \sup_{F:\mathcal{X}\times\mathcal{T}\to\mathbb{R}} \mathbb{E}_{X,T\sim P(X,T)}[F(X,T)] - \log\left(\mathbb{E}_{X,T\sim P(X)\times P(T)}\left[e^{F(X,T)}\right]\right), \tag{6}$$

where $F$ can be any measurable function from $\mathcal{X}\times\mathcal{T}\to\mathbb{R}$ that satisfies the necessary integrability constraints of two expectations in equation 6 to be well-defined, and the supremum is taken over all functions $F$ that contribute meaningfully to the optimization problem, ensuring the validity of equation 6.

To compute $I(X;T)$ in practice, assuming $n$ independent and identically distributed samples (i.i.d.) are drawn from $(X_i, T_i) \sim P(X,T)$, and $(X_i, \tilde{T}_i)$ is artificially contstructed by choosing $\tilde{T}_i$ as a randomly shuffled set of $(T_i)_{i=1}^n$. When $n$ is large enough, inspired by the law of large numbers, the MINE estimator approximates the MI as:

$$\widehat{I}_{\text{MINE}}(X;T) = \sup_{\theta\in\Theta} \frac{1}{n}\sum_{i=1}^{n} F_\theta\left(X_i, T_i\right) - \log\left(\frac{1}{n}\sum_{i=1}^{n} e^{F_\theta\left(X_i, \tilde{T}_i\right)}\right), \tag{7}$$

where $F_\theta : \mathcal{X}\times\mathcal{T}\to\mathbb{R}$ is parameterized by a deep neural network with parameters $\theta\in\Theta$, and $\Theta$ is the parameter space of the neural network. By training a neural network to optimize the above equation (i.e., finding the optimal $F_\theta$), the final output will yield the MINE estimates of MI between $X$ and $T$.

## 2.3 Motivation for Improving MI Estimation

Estimating mutual information (MI) is fundamental to various domains, ranging from learning theory to practical applications such as medical analysis and wireless communication (Shwartz-Ziv & Tishby, 2017; Saxe et al., 2018). To motivate our proposed normalization strategy, this section outlines several desirable properties that effective MI estimators should possess. Let $S = \{(X_1, T_1), (X_2, T_2), ..., (X_n, T_n)\}$ be the sampled data. With this, let $\widehat{I}^n_{est}(X;T)$ represent an estimate of the MI between $X$ and $T$ using the estimator $est$, given $N$ sampled points from the joint distribution $P(X,T)$. Ideally, we seek the estimator to have the following properties:

1. **Global Scale Invariance:** For any arbitrary $\alpha \in \mathbb{R}$ and $n \in \mathbb{Z}^+$, $\widehat{I}_{est}^n(\alpha X; \alpha T) = \widehat{I}_{est}^n(X; T)$
2. **One-Sided Scale Invariance** For any arbitrary $\alpha \in \mathbb{R}$ and $n \in \mathbb{Z}^+$, $\widehat{I}_{est}^n(X; \alpha T) = \widehat{I}_{est}^n(X; T)$

We emphasize the importance of these properties because true mutual information inherently satisfies them. By definition, $I(\alpha X; \alpha T) = I(X; T)$ and $I(\alpha X; T) = I(X; T)$ for a scalar $\alpha$. In the case of neural networks, where $X$ represents the input, $Y$ represents the target, and $T$ represents the features, estimation of $I(X; T)$ becomes important, as it was hypothesized that it can predict the generalization behaviour of deep learning networks (Shwartz-Ziv & Tishby, 2017). Furthermore, (Shwartz-Ziv & Tishby, 2017) also predicts a two-phase behaviour of $I(X; T)$ during training: (a) fitting, where $I(X; T)$ and $I(T; Y)$ increases, and (b) compression where $I(X; T)$ decreases. However, this is often not observed (Saxe et al., 2018). We hypothesize that it could be because of the scale-sensitivity of the estimators, as the scale of $T$ changes significantly during training.

We note that the current estimators may not obey one-sided scale invariance. First, we study three estimators theoretically: KSG, MINE, and binning.

## 3  Testing One-sided Scale-Invariance of MI Estimators

In this section, we theoretically test whether the common MI estimators are global-scale invariant and one-sided scale invariant. In Section 5, we also present an experimental test of one-sided scale-invariance on MI estimators. Note that for all results that follow, we assume every random variable is bounded. That is, if $X$ is bounded, we have that $|X| \leq B$ for some finite $B < \infty$. Also, for the following results, let $X \in \mathbb{R}^d$ and $T \in \mathbb{R}^m$. Note that although, some of the following results show that the estimators do not adhere to scale-invariant behaviour in the asymptotic regime, we eventually find that most estimators show significant disruption in response to scale changes of less than 10 times in magnitude.

**Binning:** Let us denote the binning estimator described in (Paninski, 2003) by $\widehat{I}_{bin}^n$. Then we have the following result.

**Proposition 1.** It holds that $\widehat{I}_{bin}^n(\alpha X; \alpha T) = \widehat{I}_{bin}^n(X; T)$ and $\widehat{I}_{bin}^n(X; \alpha T) = \widehat{I}_{bin}^n(X; T) \ \forall \alpha \in \mathbb{R}^+$.

**Remark 1.** We note that even though the binning estimator is scale-invariant, it is not a good estimator for MI, more so in the high dimension setting (Kraskov et al., 2004). This is because in high dimensions the data occupies the space very sparsely, and most bins will yield zero datapoints and thus a zero probability. Due to this, it is common practice to use fewer bins overall, which instead leads to less accurate estimates of MI as more information is lost.

**KSG:** Let us denote the KSG estimator proposed in (Kraskov et al., 2004) by $\widehat{I}_{KSG}^n$. Then, we have the following results.

**Proposition 2.** It holds that $\widehat{I}_{KSG}^n(\alpha X; \alpha T) = \widehat{I}_{KSG}^n(X; T)$, $\forall \alpha \in \mathbb{R}^+$.

This proof also leads to the following result which states that one-sided scale invariance is not a property of the KSG estimator.

**Proposition 3.** It holds that $\lim_{\alpha, n \to \infty} \widehat{I}_{KSG}^n(X; \alpha T) = -\frac{1}{k}$ and $\lim_{\alpha \to 0^+, n \to \infty} \widehat{I}_{KSG}^n(X; \alpha T) = -\frac{1}{k}$, where $k$ is the k-nearest neighbor parameter for the estimator. Thus, $\widehat{I}_{KSG}^n(X; \alpha T)$ need not be equal to $\widehat{I}_{KSG}^n(X; T)$.

**MINE:** We first define two variants of the MINE estimator as follows:

**MINE-Opt:** This estimator refers to the MINE estimator where instead of training the neural network on the loss function defined in equation 6 by stochastic gradient descent (SGD), we pick the best neural network configuration that directly maximizes equation 6. Thus, we pick the global optimum.

**MINE-SGD:** This estimator refers to the MINE estimator where optimization of the loss function defined in equation 6, is performed using conventional stochastic gradient descent. This is the standard approach proposed originally by (Belghazi et al., 2018).

We denote the MINE-based MI estimators by $\widehat{I}^n_{MINE-opt}$ and $\widehat{I}^n_{MINE-sgd}$. We then have the following results.

**Proposition 4.** It holds that $\widehat{I}^n_{MINE-opt}(X;\alpha T) = \widehat{I}^n_{MINE-opt}(X;T) \ \forall \alpha \in \mathbb{R}^+$, where $\mathbb{R}^+ = \{x \in \mathbb{R} | x > 0\}$.

Next, we outline a theoretical result regarding the limiting behaviour of the first layer weights for the MINE estimator's neural network, when the scale of one of the variables approaches zero.

**Proposition 5.** Consider the MINE optimization problem with input data $S = \{(\alpha X_1, Y_1), ..., (\alpha X_n, Y_n)\}$ where $X \in \mathbb{R}^{d_x}$, $Y \in \mathbb{R}^{d_y}$, $(X, Y) \sim P(X, Y)$ are bounded RVs and $\alpha \in \mathbb{R}^+$ is a scaling factor. We consider a neural network of depth $d_n + 1$ having $h_1, h_2, .., h_{d_n}$ relu-activated hidden neurons in the respective layers. The network is trained via stochastic gradient descent on the MINE loss function in equation 7 for a finite number of epochs $n_e$. Let the *trained* weights between the $j^{th}$ node of the $l + 1^{th}$ hidden layer and the $i^{th}$ node of the $l^{th}$ hidden layer be denoted by $w^l_{ji} \in \mathbb{R}^d$. We consider the case where the initialized weights are very close to zero but not exactly zero (to allow unsymmetrical learning). We assume that the network weights are bounded, such that every weight $|w^l_{ji}| \leq B$ for some $B \in \mathbb{R}$. Let $\eta(t)$ denote the learning rate used at epoch $t$. Then we have, $\forall i, j$,

$$\lim_{\alpha \to 0^+} |w^1_{ji}| = 0 \tag{8}$$

With this, we have the following result that explores scale invariance in neural network based MINE estimators.

**Proposition 6.** We consider the same setting as Proposition 5 for the MINE estimation problem. There, it holds that $\lim_{\alpha \to 0} \widehat{I}^n_{MINE-sgd}(X;\alpha T) = 0$. Thus, $\widehat{I}^n_{MINE-sgd}(X;\alpha T)$ need not be equal to $\widehat{I}^n_{MINE-sgd}(X;T)$.

## 4 Methodology

### 4.1 Normalization Strategies

We consider a setting where we are given an RV $X = [x_1, x_2, \ldots, x_d] \in \mathbb{R}^d$, where $x_i \in \mathbb{R}$ represents the $i$-th component of $X$. Suppose $X \sim P$, where $P$ is a probability distribution, and let $S = \{X_1, X_2, \ldots, X_n\}$ be an independent and identically distributed (i.i.d.) sample drawn from $P$, with $X_j \in \mathbb{R}^d$ for $j = 1, \ldots, n$. With this, we outline three normalization strategies that form the basis of our studies in this work. We define them as follows.

**Definition 1. (Local Normalization)** The *locally normalized variable* $X_{\sigma|S} = [x'_1, \ldots, x'_i, \ldots, x'_d] \in \mathbb{R}^d$ is defined by normalizing each dimension $i$ individually as $x'_i = \frac{x_i}{\sqrt{\mathbb{E}_S\left[(x_i - \mathbb{E}_S[x_i])^2\right]}}$, for $i = 1, \ldots, d$, where $\mathbb{E}_S[\cdot]$ denotes the empirical expectation over $S$.

**Definition 2. (Global Normalization)**
The *globally normalized variable* $X_{\Sigma|S} \in \mathbb{R}^d$ is defined as $X_{\Sigma|S} = \frac{X}{\sqrt{\mathbb{E}_S\left[\|X - \mathbb{E}_S[X]\|_2^2\right]}}$, where $\|\cdot\|_2$ denotes the $L_2$-norm, and $\mathbb{E}_S[\cdot]$ is the empirical expectation over $S$.

**Definition 3. (Global $L_\infty$ Normalization)** The *globally $L_\infty$-normalized variable* $X_{\Sigma_\infty|S} \in \mathbb{R}^d$ is $X_{\Sigma_\infty|S} = \frac{X}{\mathbb{E}_S[\|X - \mathbb{E}_S[X]\|_\infty]}$, where $\|\cdot\|_\infty$ denotes the $L_\infty$-norm and $\mathbb{E}_S[\cdot]$ is the empirical expectation over $S$.

Note that for any RV $X$, we denote by $X_{\sigma|S}$ and $X_{\Sigma|S}$ its locally and globally normalized versions respectively.

### 4.2 Studied Scale-Invariant Estimators

We are given the RVs $X \in \mathbb{R}^d$ and $T \in \mathbb{R}^m$, and sampled data $S = \{(X_1, T_1), (X_2, T_2), ..., (X_n, T_n)\} \sim P^n_{XT}$. All following estimates are for the MI between $X$ and $T$, given $S$. With this, we propose the following normalization approaches for KSG and MINE estimators. We outline our approaches for scale-invariant KSG and MINE extensions in Table 1.

**Remark 2.** In addition to the above approaches, we compare the standard baselines of KSG and MINE. Furthermore, we also include a recent variant of KSG in our comparisons, called BI-KSG (Gao et al., 2017),

Table 1: Proposed scale-Invariant KSG and MINE variants

| KSG | MINE |
|---|---|
| **KSG-Local**: $\widehat{I}^n_{KSG}(X_{\sigma|S}; T_{\sigma|S})$ | **MINE-Local**: $\widehat{I}^n_{MINE}(X_{\sigma|S}; T_{\sigma|S})$ |
| **KSG-Global**: $\max\limits_{c\in\{c_1,c_2,..,c_n\}}\left[\widehat{I}^n_{KSG}(X_{\Sigma|S}; cT_{\Sigma|S})\right]$ | **MINE-Global**: $\widehat{I}^n_{MINE}(X_{\Sigma|S}; T_{\Sigma|S})$ |
| **KSG-Global**-$L_\infty$: $\max\limits_{c\in\{c_1,c_2,..,c_n\}}\left[\widehat{I}^n_{KSG}(X_{\Sigma_\infty|S}; cT_{\Sigma_\infty|S})\right]$ | **MINE-Global-Corrected**: $\widehat{I}^n_{MINE}(\sqrt{d_X}X_{\Sigma|S}; \sqrt{d_T}T_{\Sigma|S})$ |

which has smaller bias levels for highly correlated data. We do not include binning-based measures in our experimental results, as we find that they fare poorly for almost all of our studied cases. Thus, we only study the KSG and MINE variants empirically in this work. Also, note that the range of multiplier scales $c_1, c_2, .., c_n$ are tunable hyperparameters, and we fix $c_1 = 0.1, c_2 = 0.2, ..., c_n = 2$ for all our experiments. Note that for the KSG-Global variants, this does slow down the MI estimation process, as it takes $n$ times the computation time to estimate the measure when compared to KSG.

## 5 Experimental Studies

Our empirical studies can be categorized into roughly four broad sections:

1. **Empirical motivation for proposed normalization variants:** We provide in-depth empirical analyses for each normalization variant proposed in this work, and also the overall reasons for potentially choosing global normalization approaches over local.
2. **Scale dependence and Signal to Noise Ratio (SNR) analysis of estimators:** We perform some basic tests and analyses of all estimators. First, we study their overall responses to scale changes, and then we study their responses to changes in noise levels.
3. **Accuracy analysis of estimators:** We conduct an extensive accuracy-bias-correlation analysis of all estimators in two different settings where ground truth MI is known. In each setting, we generate synthetic data using a diverse set of transformations to simulate different distribution scenarios.
4. **Studying neural network training using estimators:** We study the MI dynamics of neural networks during training. Specifically, we analyze the MI between input and features and compare the trends resulting from various estimators.

For our experiments, we use two *base* distributions for generating the random variables $X$ and $T$. We refer to them in various parts of the experiments. They are as follows:

- **Correlated Gaussians:** Here, $X \in \mathbb{R}^d \sim \mathcal{N}(0, I_d)$ and $T \in \mathbb{R}^d \sim \mathcal{N}(0, I_d)$, and $\mathbb{E}[X_iT_i] = \rho$ for $1 \le i \le d$ and $\mathbb{E}[X_iT_j = 0]$ when $i \ne j$. $I_d$ denotes the identity matrix of size $d \times d$. This is a standard setting used in many prior MI estimation works.
- **Additive Gaussian Noise:** Here $X \in \mathbb{R}^d \sim \mathcal{N}(0, I_d)$ and $T = X + \epsilon$, where $\epsilon \sim \mathcal{N}(0, \sigma^2 I_d)$.

The details for our estimators are provided in **Appendix** B. We used the NPEET MI estimator toolbox for estimating KSG and KSG-based measures [1]. For MINE, we used the popular pytorch-based package [2].

### 5.1 Additional Motivation for Normalization Variants

In this section, we provide both intuitive and empirical arguments for our proposed variants in the previous section. First, we provide intuitive and empirical reasons for when and why global normalization approaches could be preferred. Next, we provide a rationale for our proposed global normalization variants for KSG and MINE estimators.

#### 5.1.1 Global over Local

We argue in this work that global normalization should be the preferred choice, especially for estimating MI of high dimensional data, such as high dimensional feature representations in neural networks. This is mostly because local normalization makes each variable equally important, which can detrimentally affect

---

[1] https://github.com/gregversteeg/NPEET  [2] https://github.com/gtegner/mine-pytorch

the k-nearest neighbor based estimation of MI in high dimensions. In the context of neural networks, where $X$ represents the inputs and $T$ the features, often $T$ is very sparse (i.e., most values are near zero and irrelevant). By scaling these irrelevant dimensions to unit variance, it can lead to worse estimates of MI.

**KSG:** To investigate the above scenario, we conduct an experiment to evaluate the MI estimation under controlled noise conditions as follows. Given two RVs $X, T \in \mathbb{R}^2$ where $T = X + \epsilon$, with $\epsilon \sim \mathcal{N}\left(0, \sigma^2 I_2\right)$ ($I_2$ is $2 \times 2$ identity matrix). Next, a series of independent RVs represented by $\boldsymbol{\epsilon} = [\epsilon_1, \epsilon_2, ...\epsilon_k]$ where $\epsilon_i \sim \mathcal{N}\left(0, \sigma'^2\right)$, were appended to the input $X$, and concatenated to become a $2 + k$ dimensional RV $X' = [X, \boldsymbol{\epsilon}]$. A critical constraint in our design is $\sigma' \ll \sigma$, which mimics the characteristic of neural networks where irrelevant variables possess significantly less energy than relevant ones. Note that this constraint preserves $I(X; T) = I(X'; T)$. By estimating $I(X'; T)$ across increasing noise dimensions $k$, we empirically compare multiple MI estimators: KSG, BI-KSG, KSG-Local, and KSG-Global-$L_\infty$ in Figure 1, as a function of the noise dimension $k$. It is clear from the figure that while KSG and KSG-Global-$L_\infty$ maintain their estimates, KSG-Local yields significantly lower esti-

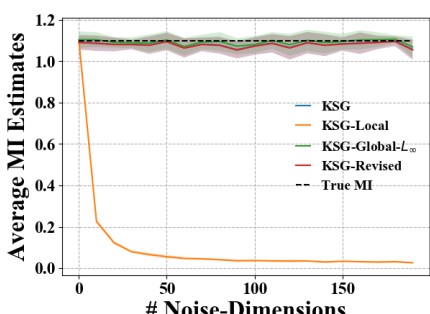

Figure 1: Average MI estimates ($I(X; T')$) for KSG-based measures for a varying number of noise dimensions ($k$). Please see Appendix A.1 for details.

mates with more noise dimensions. This is mainly due to the fact that KSG-Local will scale up the added noise variables and increase their importance.

**MINE:** To further validate our analysis, we conduct a complementary experiment investigating the robustness of MINE variants under the introduction of irrelevant noise dimensions. By employing correlated Gaussian RVs in the following experiment, we explore how different normalization variants perform across a different distribution base instead of additive noise. Specifically, we generate correlated Gaussian RVs $X, T \in \mathbb{R}^2$ with a correlation coefficient $\rho$ randomly sampled from a uniform distribution over $(0, 0.8)$ for each trial. Similar to the KSG experiment, we appended noise variables $\boldsymbol{\epsilon} = [\epsilon_1, \epsilon_2, \ldots, \epsilon_k]$ where $\epsilon_i \sim \mathcal{N}\left(0, \sigma'^2\right)$ to $X$ to generate $X' = [X, \boldsymbol{\epsilon}]$. The experiment computes the average bias of all MINE variants across 10 trials for each choice of $k$. In each trial, we choose a random correlation co-efficient $\rho$. This approach enables a comprehensive examination of the average bias of all estimators across a broad range of correlations.

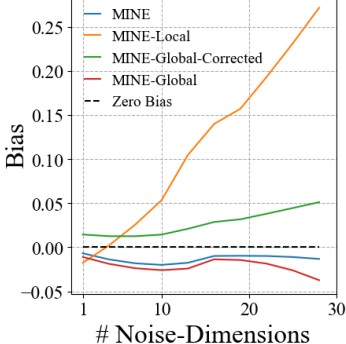

Figure 2: Bias of MINE-based measures for varying noise dimensions. Please see Appendix A.1 for details.

Intuitively, we might expect similar behavior as the KSG experiment, as the noise dimensionality $k$ increases, the estimates of MI should reduce and the bias should become increasingly negative. However, our findings present a counterintuitive phenomenon. As summarized in Figure 2, local normalization dramatically affects the bias positively rather than negatively in the case of MINE-Local.

Specifically, Figure 2 shows that as more noise dimensions are appended, the MINE-Local estimates tend to grow significantly beyond the true MI, whereas the other measures, including the MINE-Global variants, remain stable. Our explanation is as follows. Unlike KSG, for MINE there is a neural network that actively seeks to maximize the DV objective. It is well known in the literature that neural networks can fit random noise data very well Zhang et al. (2017). Furthermore, as noise dimension $k$ increases, the overall data dimension increases and so does the number of network parameters, which enables the network to maximize the DV objective better. For MINE, we do not see an increase because the added variables are of very low energy, and thus the network's *effective* input dimensionality does not change as the added noise variables have a negligible impact on the output of the network.

### 5.1.2 KSG-Global: Why the Maximization Step?

We outline two main arguments behind the maximization step for KSG variants in Table 1.

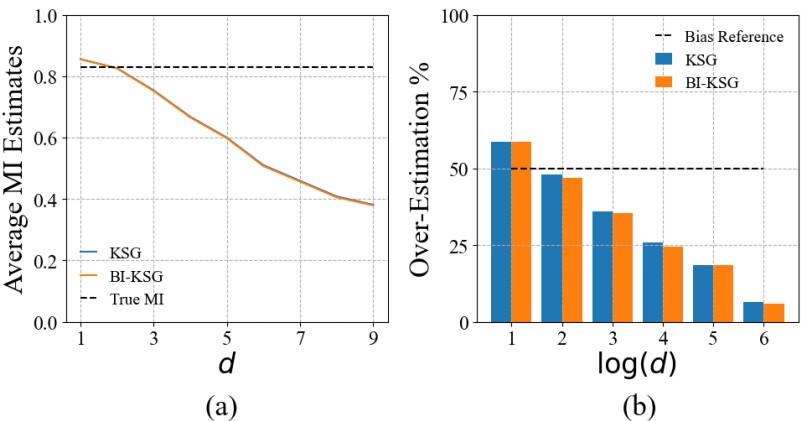

Figure 3: Analyzing the dependency of the bias of both KSG estimators with the data dimension. Please note that log here is in base 2. Please see Appendix A.1 for details.

1. **Negative bias in high dimensions:** We find that the KSG estimator has a significant negative bias for data in high dimensions, echoing similar observations in Czyż et al. (2023). More specifically, we find that the bias of the KSG estimator grows significantly with data dimension in the negative direction. We show this in two ways. First, we conduct an experiment where $X \in \mathbb{R}^d$ and $T \in \mathbb{R}^d$ are correlated Gaussians with a correlation coefficient $\rho$ chosen such that the ground truth MI is fixed at a certain value (around 0.8). The dimensionality $d$ is increased from 1 to 9 in steps of one. We chose to fix the ground truth MI across dimensions, as otherwise average MI would grow with data dimension, and we did not want the negative bias to be a result of ground truth MI growing faster than the KSG estimates. For each $d$, we run 20 trials, where in each trial 1000 data points were sampled from the joint distribution $P(X, T)$. We record KSG's average estimate of MI for each $d$, and the results are shown in Figure 3(a). It is clearly evident that the KSG estimate has a growing negative bias with dimensionality in this scenario.

To get a more general idea of the trends of the bias of the KSG estimator in response to increasing data dimension, we conduct another experiment. Here, same as before, $X \in \mathbb{R}^d$ and $T \in \mathbb{R}^d$ are correlated Gaussians. However, the correlation coefficient $\rho$ is randomly chosen from a pre-determined range. Furthermore, we conduct 100 trials for each choice of dimensionality $d$, and in each trial, the estimator has access to 200 sampled datapoints from $P(X, T)$. We choose a broader range of dimensions, such that $d = [2, 4, 8, 16, 32, 64]$. Lastly, we plot the percentage of trials in which the estimated MI was greater than or equal to the ground truth MI. The results are shown in Figure 3(b). Note that for both KSG and BI-KSG, the proportion of samples where the estimated MI was lower than the ground truth MI increases significantly as $d$ increases. To that end, we see that when $d = 64$, most estimates of MI are strictly less than the ground truth values. These two studies indicate that taking the maximum of multiple estimates of MI from KSG can potentially reduce the negative bias and improve accuracy, especially for high-dimensional data. After all, if $\widehat{I}_{KSG}^n(X_{\Sigma|S}, cT_{\Sigma|S})$ is always less than the true MI irrespective of $c$, the maximum value in these cases will always have the least bias. This is supported by our empirical results in Tables 2 and 3.

2. **Consequence of Proposition 3:** Proposition 3 finds that the KSG estimator for $I(X; \alpha T)$ converges to a negative value at either end of the scale spectrum w.r.t $\alpha$. This motivated us to consider the maximum estimate of MI $\widehat{I}_{KSG}^n(X_{\Sigma|S}, cT_{\Sigma|S})$ across a range of scales in $c$. Note that as both $X_{\Sigma|S}$ and $T_{\Sigma|S}$ represent the global normalized versions of $X$ and $T$, we can fix this pre-determined range of scales in $c$. We later see that the $\widehat{I}_{KSG}^n(X_{\Sigma|S}, cT_{\Sigma|S})$ follows an almost Gaussian like trend w.r.t $c$ (Figure 6a).

**Remark 3.** Note that MINE has an implicit maximization over relative scales in the way it is optimized. This is mainly because the weights of the first layer can be any arbitrary set of real numbers as per the optimization objective. Furthermore, scaling the weights associated with one of the input RVs $X$ or $T$ is equivalent to scaling $X$ or $T$ respectively, as $(\alpha W)^T X = W^T(\alpha X)$. It is important to note that the maximization goes beyond just relative scales though, as the network function should ideally be invariant to affine transformations of the input. This suggests that MINE intrinsically considers a maximization of MI over all affine transformations of both variables. However, due to the nature of the gradient descent

approach used to optimize the network, and its preference for flatter minima Keskar et al. (2017), this may not materialize to the fullest.

### 5.1.3 Motivation for Global Normalization Variants

In this section, we provide motivation for the specific global normalization variants proposed in this work: KSG-Global-$L_\infty$ and MINE-Global-Corrected.

**KSG:** One of the objectives of global normalization is to put both RVs $X$ and $T$ on equal footing w.r.t nearest neighbor distances, such that the KSG estimator is not biased towards any one variable, which leads to low and potentially even negative estimates (Proposition 3 and Figure 6a). In contrast, local normalization puts every dimension of $X$ and $T$ on an equal footing, which risks amplifying the impact of noisy and irrelevant dimensions, as demonstrated before. However, it is important to consider that KSG's nearest neighbor distances are computed using the $L_\infty$-norm, instead of $L_2$-norm. Therefore, it is possible that KSG-Global may not put $X$ and $T$ on an equal footing w.r.t nearest neighbor distances. In fact, when $d_X \gg d_T$, the $L_\infty$-nearest neighbor distances for $X_{\Sigma|S}$ will be significantly lower than for $T_{\Sigma|S}$. This is because global normalization will ensure that the average

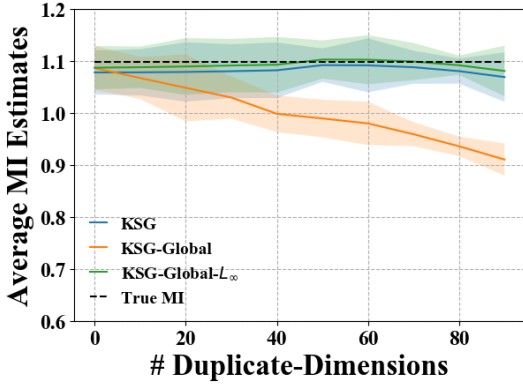

Figure 4: Data duplication: Average MI estimates for KSG-based approaches. Please see Appendix A.1 for details.

energy of dimensions sum to 1, and as $d_X \gg d_T$, the scale of individual dimensions in $X$ will be significantly smaller than in $T$. As $L_\infty$-norm only considers the largest element in the vector, this implies that $L_\infty$ distances of $X_{\Sigma|S}$ can turn out much smaller than for $T_{\Sigma|S}$ in this case.

To illustrate this, we conduct an experiment, following the same setup as before with the Gaussian noise addition dataset. We have $X, T \in \mathbb{R}^2$ where $T = X + \epsilon$ and $\epsilon \sim \mathcal{N}(0, \sigma^2 I_2)$. We then increase the dimensionality of $X$ by simply appending a number of its duplicates to yield $X' = [X, X, X, \ldots, X](k$ times). This ensures that we preserve the distance structure of $X$ in $X'$. Let the number of duplicate copies be denoted by $k$. Note that $I(X'; T) = I(X; T)$. For every $k$ we undergo 10 trials, and in every trial we sample 200 data points from $P(X, T)$ and obtain MI estimates of KSG and KSG-Global variants. The results are shown in Figure 4. As hypothesized, we see that KSG-Global shows a clear reduction as $k$ increases. In contrast, both KSG and KSG-Global-$L_\infty$ are steady and have roughly consistent average MI estimates. This shows the importance of using $L_\infty$-norm to estimate distances instead of $L_2$ in the case of KSG, as it uses $L_\infty$-norm for estimating nearest neighbor distances. Next, we discuss the MINE variants.

**MINE:** As discussed in KSG's case, global normalization can yield low individual energy per dimension if the data dimensionality is large. In the case of MINE, if $d_X \gg d_T$, we will have that $\mathbb{E}_i[X_{\Sigma|S}(i)^2] \ll \mathbb{E}_i[T_{\Sigma|S}(i)^2]$, where $X_i$ and $T_i$ denote individual dimensions of $X$ and $T$ respectively. In fact, $\mathbb{E}_i[X_{\Sigma|S}(i)^2] = \frac{1}{d_X}$ and $\mathbb{E}_i[T_{\Sigma|S}(i)^2] = \frac{1}{d_T}$. From the perspective of gradient descent and backpropagation, this implies that most error signals will focus on $T$, and $X$ will be relatively neglected. Furthermore, if both $d_X$ and $d_T$ are large, the network input will have low energy per dimension, which may affect the optimization adversely. So, to avoid this, we rescale the global normalized data $X_{\Sigma|S}$ and $T_{\Sigma|S}$ to $X'_{\Sigma|S}$ and $T'_{\Sigma|S}$, such that the average energy of every dimension $\mathbb{E}[X'_{\Sigma|S}(i)^2] = \mathbb{E}[T'_{\Sigma|S}(j)^2] = 1$, $\forall i, j$. Thus, $X'_{\Sigma|S} = \sqrt{d_X} X_{\Sigma|S}$ and similarly $T'_{\Sigma|S} = \sqrt{d_T} X_{\Sigma|S}$, which yields the MINE-Global-Corrected approach. Note that rescaling still preserves the relative energies between different dimensions, i.e., $\mathbb{E}[X(i)^2]/\mathbb{E}[X(j)^2] = \mathbb{E}[X'_{\Sigma|S}(i)^2]/\mathbb{E}[X'_{\Sigma|S}(j)^2]$.

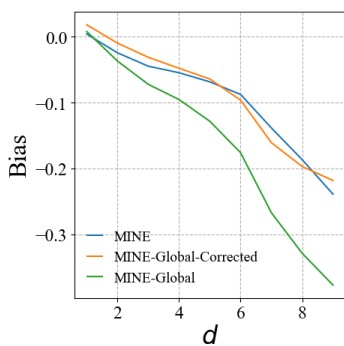

Figure 5: Estimator bias versus dimension: Comparing MINE with MINE-Global variants. Please see Appendix A.1 for details.

To showcase the importance of rescaling the globally normalized variables, we conduct an experiment where $X, T \in \mathbb{R}^d$ are correlated Gaussian variables with a correlation coefficient of $\rho$ between the corresponding dimensions of $X$ and $T$. Like in previous experiments, $\rho$ is chosen randomly from a specified range for each trial. We vary the dimensionality $d$ from 1 to 9, and for each $d$ we conduct ten trials. In each trial, we generate 1000 data points from $P(X, T)$, and compare MINE estimates with its global normalization variants. For every $d$, we ultimately compute the average bias of each estimator. Results are shown in Figure 5. Our observations are two-fold. First, we observe that in general MINE estimates also yield a growing negative bias with larger input dimensionality. Next, we observe that MINE-Global grows negative at a faster rate compared to MINE, but MINE-Global-Corrected shows similar bias trends as MINE. The results imply that when the input signals are low due to global normalization, MINE tends to yield lower estimates on average. Also, they show that our rescaling approach is able to address this and yield similar bias levels as the original MINE.

### 5.1.4 Scale and SNR analysis

**Scale:** We conduct two sets of experiments. First, we generate $X, T$ using a correlated Gaussian base. Then, we generate datasets following $P(X, T)$ across 20 trials. In each trial, we generate 1000 samples from $P(X, T)$. Using this set of 20 datasets, we construct many other copies of this set by scaling $X' = \eta X$, where $\eta$ represents the scaling factor. We choose 20 different $\eta$ between $10^{-2}$ and $10^3$ for KSG, and between $10^{-2}$ and 10 for MINE, such that they are equispaced in a $\log_{10}$ scale. We choose different ranges for KSG and MINE, because the MINE estimates fall rapidly around $\eta = 10$ and yield highly negative values after that. For each $\eta$, we compute the average values of the estimators across the 20 datasets and report the average estimates as a function of $\eta$ in Figures 6a and 6b. This concludes the first part of our experiments.

Next, we analyze the degree of estimation error, as the scale of $X$ varies via $\eta$. To get a robust measure of error, we conduct 20 trials for every choice of $\eta$, and in each trial, we sample $X, T$ from a correlated Gaussian base and set $\rho$ randomly within a specified interval. After sampling $X, T$, we generate $X', T$ by scaling $X$, as $X' = \eta X$. We then measure the root-mean-squared error (RMSE) between the MI estimates and the ground truth MI across the 20 trials and repeat the process for every choice of $\eta$. To get more general trends of

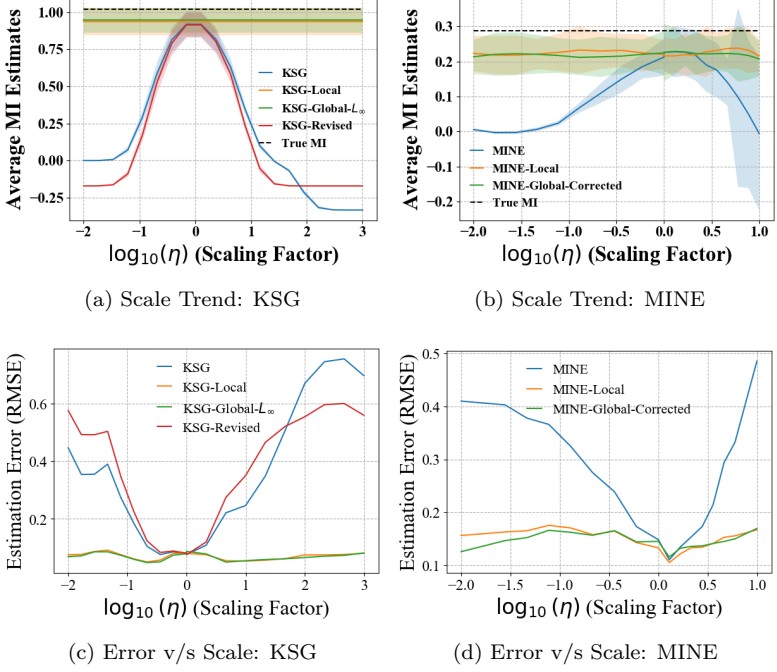

(a) Scale Trend: KSG

(b) Scale Trend: MINE

(c) Error v/s Scale: KSG

(d) Error v/s Scale: MINE

Figure 6: Analysis of MI Estimators in response to data scaling. Estimates are for $I(\eta X; T)$, where $\eta$ is the scaling factor. Please see Appendix A.1 for details.

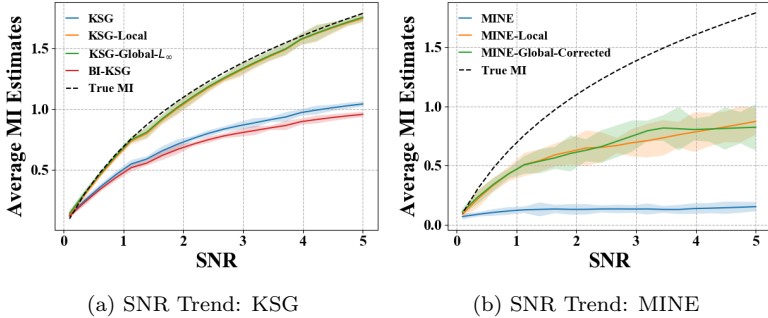

(a) SNR Trend: KSG

(b) SNR Trend: MINE

Figure 7: Average MI estimates for various estimators across different values of SNR. Estimates are for $I(X; \eta T)$, where $T'$ is generated by adding Gaussian noise to $X$, the level of which controls the SNR. We fix $\eta = 0.1$. Please see Appendix A.1 for details.

error, for every choice of $\eta$, and for every trial, we choose to sample $\rho$ randomly. This gives a wider range of ground truth MI. We plot the RMSE values for every measure for every scale factor $\eta$ in Figures 6c and 6d.

**Takeaways:** We see that the standard MI estimators for KSG and MINE are significantly affected by scale. Furthermore, we see that KSG estimates converge to very low and even negative values as $\eta$ reaches either extreme. Interestingly, as $\eta$ grows, we find that KSG estimates indeed converge to around $-0.33$ which is $1/k$ as $k = 3$ for our experiments. This validates the result in Proposition 3. However, on the other side, when $\eta$ reduces to very small values, we find that the estimates reach zero. This is because the estimator we used has a small distance correction in its k-nearest neighbor search. When we remove that correction, we find that the estimates converge to $-0.33$ for both extremes of $\eta$. We also see that MINE estimates converge to zero as $\eta$ reduces. This validates our result in Proposition 6. For both KSG and MINE, we see that the local and global variants stay robust in terms of scale. For KSG, the values stay essentially level as $\eta$ varies, but for MINE there are some small fluctuations. For both cases, we see that the local and global variants show significantly less estimation error as the scaling factor diverges from one. Lastly, we also see that KSG overall has much tighter confidence intervals than MINE. Most notably, when the scaling factor increases, MINE's estimates show significantly higher variance and become unstable.

**SNR:** In this section, we show how the scale dependence of MI Estimators can lead to other scenarios where they exhibit trends that are not ideal. We consider $X, T$ sampled from the additive Gaussian noise base. Thus, we have $T = X + \epsilon$, where $\epsilon \sim \mathcal{N}(0, \sigma^2)$. We additionally scale $T$ to obtain $T' = 0.1T$. We scale down $T$ to $T'$ so that the scale-dependent bias of MI estimators becomes a factor in our experiments. Note that $I(X; T) = I(X; T')$, and thus when $\sigma^2$ increases, $I(X; T)$ should continue to decrease and vice versa. We vary $\sigma^2$ such that the Signal-to-noise ratio (SNR) ranges between 0 and 5. For every choice of SNR, we conduct 10 trials. In each trial, we generate 1000 samples of $X, T'$ according to $P(X, T')$. Lastly, we average the MI estimates for each estimator across the trials. The process is repeated for all values of SNR in this range. Results are shown in Figures 7a and 7b.

**Takeaways:** We first note that the global and local variants of the measures follow a similar trend compared to the ground truth MI; the average MI estimates grow with SNR. However, interestingly, we see that for the vanilla KSG and MINE estimators, their average MI estimates stop growing after a while and seem to converge. This shows that the scale-dependence of unnormalized estimators can potentially yield incorrect trends of true MI in other settings where scale of the variables can confound the true MI.

## 5.2 Comparing MI Estimators: Error Analysis

In this section, we undergo a comprehensive series of experiments, where we compute various error measures of all estimators on a diverse range of datasets.

Table 2: Normalized RMSE of KSG-Based Estimators: Additive Gaussian Noise Base

| Transformation | | | | | $d$ | KSG-Based Measures | | | | |
|---|---|---|---|---|---|---|---|---|---|---|
| rm | cb | sg | ds | dn | | ksg | bi-ksg | ksg-loc | ksg-glo | ksg-glo-$L_\infty$ |
| | ✓ | ✓ | ✓ | | 2 | 0.351 | 0.404 | 0.147 | 0.208 | **0.110** |
| | | ✓ | ✓ | | 2 | 0.286 | 0.335 | 0.060 | 0.084 | **0.050** |
| | ✓ | ✓ | | | 2 | 0.458 | 0.533 | 0.145 | **0.113** | 0.113 |
| | | ✓ | | | 4 | 1.275 | 1.424 | 0.312 | **0.300** | 0.301 |
| ✓ | | | | ✓ | 4 | 0.862 | 0.932 | 0.445 | 0.594 | **0.396** |
| | | | ✓ | | 4 | 0.332 | 0.342 | **0.304** | 0.520 | **0.297** |
| | | | | ✓ | 4 | 0.332 | 0.342 | 1.327 | **0.298** | 0.297 |
| ✓ | ✓ | | | | 4 | 1.131 | 1.233 | 0.977 | 0.959 | **0.931** |
| | | ✓ | | ✓ | 4 | 1.275 | 1.424 | 1.334 | **0.300** | 0.301 |
| | | ✓ | ✓ | ✓ | 6 | 1.981 | 2.129 | 1.904 | 1.021 | 1.021 |
| | | ✓ | | | 6 | 1.983 | 2.131 | **0.816** | 0.811 | 0.812 |
| | | ✓ | ✓ | | 6 | 1.377 | 1.408 | 1.343 | 1.605 | 1.290 |
| ✓ | | | ✓ | | 6 | 1.643 | 1.730 | 1.275 | 1.660 | 1.153 |
| | | ✓ | | ✓ | 6 | 1.983 | 2.131 | 1.905 | **0.814** | 0.816 |

### 5.2.1 Experiment Summary

**Dataset creation:** To create these datasets, we follow the two base distributions described in the beginning of this section. First, we generate $X, T$ according to the two base distributions: Additive Gaussian and Correlated Gaussian as defined in Section 5. Then, we then make $X$ undergo some (or none) of the following transformations, which are all MI preserving. For what follows, let $X \in \mathbb{R}^d$ and $T \in \mathbb{R}^d$.

1. **Randmat (rm):** $X' = \alpha W^T X$, where $\alpha \sim Unif(0,1)$ and $W \in \mathbb{R}^{d \times d}$ where $W(i,j) \sim Unif(0,1)$. $Unif(a,b)$ denotes a uniform distribution over $[a,b]$. If the randomly generated $W$ is not invertible, we keep generating until we get an invertible $W$.
2. **Cube (cb):** $X' = X \circ X \circ X$, where $\circ$ denotes element wise multiplication (Hadamard Product).
3. **Sigmoid (sg):** $X' = \sigma(X)$, where $\sigma : \mathbb{R}^d \to \mathbb{R}^d$ is such that $X'[i] = \frac{1}{1+e^{-X[i]}}$, where $X[i]$ denotes the $i^{th}$ dimension of $X$ and similarly for $X'$.
4. **Duplicate-self (ds):** $X' = [X, X, ...X] \in \mathbb{R}^{Kd}$. We set $K = 20$ in our experiments.
5. **Duplicate-noise (dn):** $X' = [X, \boldsymbol{\epsilon}] \in \mathbb{R}^{d+k}$, where $\boldsymbol{\epsilon} = [\epsilon_1, \epsilon_2, ...\epsilon_k]$ where $\epsilon_i \sim \mathcal{N}\left(0, \sigma'^2\right)$. We set $\sigma' = 0.2$ and $k = 20$.

Our objective is to evaluate the accuracy of the estimation of $I(X'; T)$.

**Performance Measures:** We study three different measures of performance in our experiments. For what follows, let $\hat{\mu}_1, \hat{\mu}_2, ...\hat{\mu}_k$ denote the estimated values of MI for any estimator across $k$ trials, and let $\mu_1, \mu_2, ...\mu_k$ denote the ground truth values. With this, we summarize our performance measures as follows:

- **Normalized RMSE:** We first estimate the RMSE as $RMSE(\hat{\boldsymbol{\mu}}, \boldsymbol{\mu}) = \sqrt{\mathbb{E}_i[(\hat{\mu}_i - \mu_i)^2]}$. Then we estimate a basline RMSE as $RMSE\_Base(\boldsymbol{\mu}) = \sqrt{\mathbb{E}_{i,j}[(\mu_i - \mu_j)^2]}$. With this, we can estimate the final measure as: $RMSE\_Norm(\hat{\boldsymbol{\mu}}, \boldsymbol{\mu}) = \frac{RMSE(\hat{\boldsymbol{\mu}}, \boldsymbol{\mu})}{RMSE\_Base(\boldsymbol{\mu})}$.
- **Spearman Correlation:** The Spearman correlation measures the degree of monotonic relationship between $\hat{\boldsymbol{\mu}}$ and $\boldsymbol{\mu}$ (Zar, 2005). This is estimated as the Pearson's correlation coefficient between the rank values of $\hat{\boldsymbol{\mu}}$ and $\boldsymbol{\mu}$.
- **Bias:** We estimate the bias as $\mathbb{E}_i \left[ \mu_i - \hat{\mu}_i \right]$.

**Evaluation Process:** We summarize the empirical process for the results in Tables 2, 3, which are for the additive Gaussian noise distribution base, as follows. The full results, including the results for the correlated Gaussian distribution base, with the Spearman correlation and bias measures are provided in **Appendix C**. For each experiment, we consider a specific set of transformations to be applied to $X$, which is shown in the first column of the tables. Once chosen, we then undergo 40 trials of data generation and MI estimation. In each trial, we generate $N$ samples of $X, T \sim P(X, T)$ according to the base distribution, and then generate the transformed $X'$ according to the list of transformations in the corresponding row. For Table 2 and 3,

Table 3: Normalized RMSE of MINE-Based Estimators: Additive Gaussian Noise Base

| Transformation | | | | | d | MINE-Based Measures | | | |
|---|---|---|---|---|---|---|---|---|---|
| rm | cb | sg | ds | dn | | mine | mine-loc | mine-glo | mine-glo-corr |
| | ✓ | ✓ | ✓ | | 2 | 0.470 | 0.292 | 0.337 | **0.278** |
| | ✓ | ✓ | | | 2 | 0.445 | 0.255 | 0.275 | **0.233** |
| | | ✓ | ✓ | | 2 | 1.036 | **0.560** | 0.658 | **0.565** |
| | | | ✓ | | 4 | 1.302 | 0.720 | 0.968 | **0.684** |
| ✓ | | | | ✓ | 4 | 0.930 | 0.438 | 0.803 | **0.381** |
| | | ✓ | | | 4 | 0.276 | **0.369** | 0.642 | 0.375 |
| | | | | ✓ | 4 | 0.622 | 0.423 | 0.895 | **0.269** |
| ✓ | ✓ | | | | 4 | 1.574 | 1.099 | 1.219 | 1.162 |
| | | ✓ | | ✓ | 4 | 1.335 | 0.423 | 0.895 | **0.286** |
| | | ✓ | ✓ | ✓ | 6 | 1.881 | 0.570 | 1.516 | **0.437** |
| | | | ✓ | | 6 | 1.843 | 1.147 | 1.535 | 1.185 |
| | | ✓ | ✓ | | 6 | 1.213 | 0.991 | 1.444 | 1.004 |
| ✓ | | | ✓ | | 6 | 1.014 | 1.088 | 1.441 | 1.058 |
| | | ✓ | | ✓ | 6 | 1.831 | **0.517** | 1.499 | 0.545 |

we set $N = 1000$. The data dimensionality of $X$ and $T$ is represented via $d$ and is shown in the tables. After generating $N$ samples, we then obtain MI estimates ($I(X'; T)$) from all estimators. Over the course of 40 trials, we then estimate the three different performance measures outlined in the previous section for all estimators. Lastly, the red entries in the Tables refer to the case where the estimator error exceeds the base RMSE, yielding a normalized RMSE of greater than one. These results are thus not significant in terms of RMSE. However, even in these cases, we find that the MI estimates are often significantly correlated with the true MI (with Spearman correlation).

**Remark 4.** Note that as each transformation is MI preserving, we can combine them in arbitrary ways and generate completely new transformations and data distributions. As we know the ground truth MI of the original base distribution, the transformed data will also have the same ground truth MI. This framework allows us to model a flexible set of distributions, which allow us to create high dimensional data with a low dimensional intrinsic dimension, which is often the case for neural network features. To illustrate, our data dimension can reach up to 200 dimensions, with very low intrinsic dimension ($<10$), compared to the experiments in Czyż et al. (2023) which go up to 25 dimensions. Furthermore, our choice of transformations is motivated by the choice of estimators tested in this work, and the normalization strategies compared in this work. For instance, local normalization typically performs poorly with added noise variables (duplicate-noise), and the KSG-Global variant is not consistent in response to addition of duplicate dimension (duplicate-self). KSG itself is also affected by transformations such as sigmoid and cube as that can drastically change the distances in the nearest neighbor computation, and thus alter the structure of the data.

### 5.2.2 Takeaways

The main observations from the results are as follows:

- Overall, global and local normalization variants fare significantly better than the baseline measures.
- Our global normalization variants (MINE-global-corrected and KSG-Global-$L_\infty$) overall fare better than other normalization strategies. In fact when the base distribution is additive Gaussian noise, we find that in most cases MINE-global-corrected and KSG-global-$L_\infty$ outperform compared to the other normalization approaches.
- KSG-Global-$L_\infty$ has very consistent performance, and across both settings, it seems to have the best performance in most cases. Even when the normalized RMSE estimates are insignificant (red entries), KSG-Global shows significant correlation with true MI in many of the cases.
- As discussed in our motivation, we find that overall the global normalization variants (MINE-Global-Corrected and KSG-Global-$L_\infty$) perform better than their vanilla global normalization counterparts. This is much more apparent in the case of MINE.

## 5.3 Application of MI estimations in Deep Learning

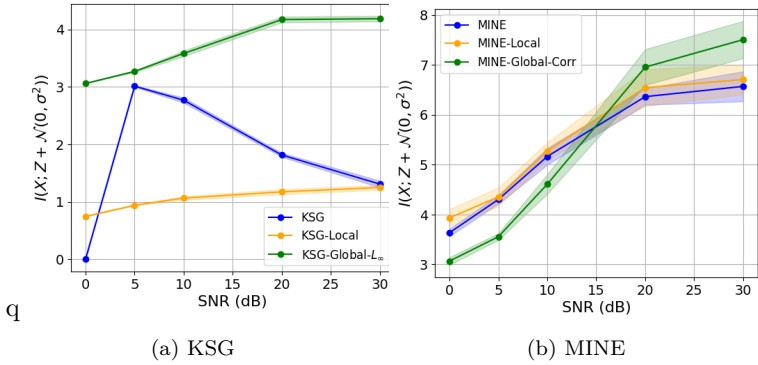

q

(a) KSG           (b) MINE

Figure 8: $I(X; Z + \mathcal{N}(0, \sigma^2))$ results with varying SNR. Shaded regions indicate 95% confidence intervals derived from 10 trials.

As an important measure of the dependence between two variables, mutual information is also widely used to analyze the behavior of neural networks during training. In this section, we present a comprehensive analysis of MI measures on network behaviors across datasets, including IB dataset (Shwartz-Ziv & Tishby, 2017), MNIST (Deng, 2012) and CIFAR-10 (Krizhevsky & Hinton, 2009). Network architecture architectures and activation functions and other details are provided in Appendix A.

The values of MI measures are obtained as follows. Given a dataset for classification $\{X, Y\}$, we train networks with $\{X, Y\}$, and during training, we extract the outputs of one selected intermediate layer, denoted as $Z$. For the IB dataset, the $Z$ is extracted from the output of the third layer of the network, For the MNIST dataset, the $Z$ is extracted from the output of the third layer of the network. For the CIFAR-10 dataset, the $Z$ is extracted from the output of the Global Average Pooling layer of the network. After obtaining $X$, $Y$ and $Z$, we analyze the network behaviors by observing the trend of two MI measures, $I(X; Z)$ and $I(Z; Y)$, with KSG, KSG-Local, MINE, MINE-Local, and proposed KSG-Global-$L_\infty$ and MINE-Global-Corrected estimators.

We analyze $I(X; Z)$ and $I(Z; Y)$ during neural network training from three perspectives:

- **Impact of noise**: We examine how the MI changes when additive Gaussian noise is introduced in the intermediate layers of the network.

- **Training dynamics**: We investigate the changes of MI estimates $I(X; Z)$ over the course of training epochs.

- **Information plane visualization**: We plot $I(X; Z)$ against $I(Z; Y)$ to visualize the information plane, providing insights into the trade-off between the information preserved about the input $X$ and the information relevant to the label $Y$. We put the results for information plane in **Appendix** E.

**Impact of noise in neural network training:** In this experiment, $I(X; Z + \mathcal{N}(0, \sigma^2))$ is measured at the last epoch of the training to evaluate how the MI changes when additive Gaussian noise is introduced. an additive noise layer $N \sim \mathcal{N}(0, \sigma^2)$ is introduced before the $Z$ layer. In figure 8, the noise level is characterized using the signal-to-noise ratio (SNR) numbers, which quantifies the strength of a signal relative to the background noise. Specifically, an SNR of $a$ dB implies that with unit signal power, the noise variance is $10^{-a/10}$. Consequently, as SNR increases, the noise level decreases, leading to a reduction in interference. Briefly, the mutual information $I(X; Z + \mathcal{N}(0, \sigma^2))$ is expected to increase as the noise level decreases. Since the SNR is expressed in decibels (dB), the noise reduction is more pronounced within the range of low SNRs (e.g. 0 to 15 dB), decreasing faster compared to the range of high SNRs (15 to 30 dB), and MI within the range of low SNR should also have a higher growing rate compared to high SNRs.

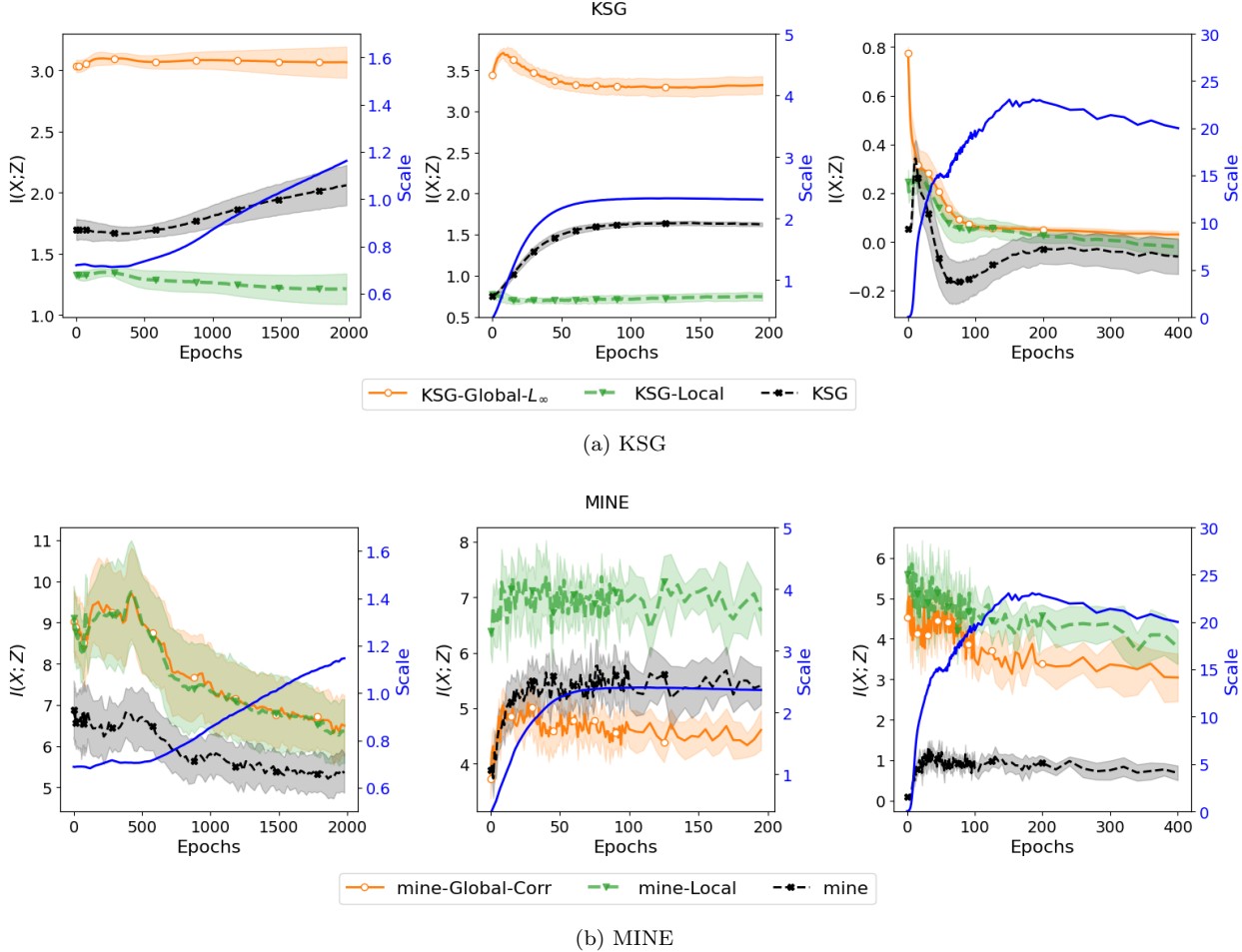

(a) KSG

(b) MINE

Figure 9: $I(X; Z)$ measures estimated after every epoch of training on IB, MNIST and CIFAR-10 datasets. $Z$ represents the output of $3^{rd}$ layer for IB dataset and MNIST dataset, and $7^{th}$ layer for CIFAR-10 dataset. Details in Appendix B.3.

In figure 8, we present the change of $I(X; Z + \mathcal{N}(0, \sigma^2))$ during training, comparing the original KSG estimator, its local-normalized and global-normalized variants, and the MINE estimator and its variants. The displayed results represent the averages from 10 trials. For the original KSG estimator, we observe that as the SNR increases, the mutual information does not change as initially anticipated. Instead, the results for the original KSG estimator initially increase and then decline as SNR continues to rise. Notably, both estimators with global normalization exhibit the most consistent trend, reflecting the expected increase in dependence between $X$ and $Z$ as noise is reduced, and have a higher growing rate at low SNR regime.

$I(X; Z)$ **versus the number of epochs:** As we have already established that the KSG estimator is sensitive to data scale, we wanted to see to what extent this is the case during the training of neural networks. In figure 9, we present the change of $I(X; Z)$ during training, comparing the original KSG estimator, its local-normalized and global-normalized variants, and the MINE estimator and its variants. The displayed results represent the averages from 10 trials. We also simultaneously plot the scale of the features $Z$ (i.e., $|Z|$), and its y-scale is placed on the right side of the figures. We find that in MNIST and CIFAR-10 datasets, $I(X; Z)$ from the original KSG estimator shows a significantly high correlation with the scale of the features during training, which hints that it may fundamentally capture the changes in feature scale. In contrast, the global-normalized estimate does not follow the scale curve and yields interesting trends. For the IB and CIFAR-10 datasets, the global-normalized estimate first increases and then decreases after training a

certain number of epochs, thus being more adherent to the original fitting followed by the compression trend proposed by (Shwartz-Ziv & Tishby, 2017). In CIFAR-10, the decrease in $I(X;Z)$ happens right after 3 epochs, which demonstrates a completely different trend than the baseline measures. Overall, for KSG, we note that in two of the three cases we see a clear fitting and compression phase as described in (Shwartz-Ziv & Tishby, 2017) using KSG-global-$L_\infty$, which is not the case for other variants.

In figure 9, we find that the MI estimates obtained by MINE estimators are noisier. This may imply that MINE estimators may face challenges when dealing with high-dimensional variables that do not follow a standard distribution, particularly when the sample size is relatively small (N = 5000). However, we do observe fitting and compression for both local and global variants in IB and MNIST. In (Poole et al., 2019), authors highlighted that MINE often shows higher variance due to neural network training instability.

Lastly, we also perform the information plane analysis, which we report in Appendix E.

## 6 Discussions on Other Related Works

### 6.1 Equitability

There is a body of work on the equitability of mutual information estimators, of which the notion of self-consistent equitability is related to our work. It seems (Reshef et al., 2013) is the first that proposes the notion of self-consistent equitability, which is a generalized form of one-sided scale invariance discussed in our work, as we consider the specific case when the function is scale-related. In our work, however, we do not focus on the aspect of data transformation specifically, but mainly focus on the behaviour of estimators in response to scaling. In such works concerned with self-consistent equitability, there is always a very wide range of functions (Table 2 of (Reshef et al., 2013)) to get a broad overview of the behavior of MI in response to any type of transformation. However, it is notable that scaling has not been one of them, most likely due to the way the data was already preprocessed to be scale-invariant, using local normalization, which already makes them self-equitable to scaling.

Our work only indirectly tests self-consistent equitability in Tables 1 and 2, where instead of prioritizing the degree of self-consistent equitability, we prioritize the overall MI estimation accuracy. We also use a variety of transformations as functions on the data (in a cascaded manner) and see the resulting estimation errors for the various compared metrics. However, our choice of transformations (apart from cube and sigmoid) is very different when compared to Table 2 of Reshef et al. (2013)'s work, as they either encompass all dimensions (like invertible random matrix multiplication) or change the dimensionality of the input by adding noisy dimensions or duplicate dimensions. In contrast, most functions in self-consistent equitability literature work with one-dimensional transformations applied to each data dimension individually. Lastly, as we cascade a random subset of these operations in a random order, we get a richer set of MI-preserving transformations, which can potentially also be tested in an equitability setup using their choice of metrics (like in Figure 2 of (Kinney & Atwal, 2014)). We are considering this for future work.

### 6.2 Recent MI Estimation works

We outline three different aspects in which our work differs from (Czyż et al., 2023), which is an important recent study that also tests MI Estimators and their performance under various settings.

**Choice of transformations:** As the focus in (Czyż et al., 2023) is not particularly on the neural network use case where $X$ is the input and $Z$ represents a feature layer, their choice of transformations is motivated differently. Most of their transformations are dimension-wise, i.e., a transformation applied to each dimension. The only exception is their spiral diffeomorphism (Figure 5 of (Czyż et al., 2023)), which radially morphs the distribution in such a way that the MI is preserved. We note that as our focus is mainly on the natural use cases of MI in deep learning, we construct certain types of transformations relevant to this setting. Apart from the cubic transformation, which is dimension-wise, all our other transformations are motivated by the potential use case in deep learning. We outline each one as follows. The sigmoid transformation is motivated by potential uses of sigmoid in the network's hidden layers. The random matrix multiplication (randmat) transformation is motivated via the features undergoing similar transformations

through neural network layers, which are usually matrix multiplications followed by non-linearity. Note that the randmat also has an additional scaling term $\alpha$ (Section 5.2), which scales the resulting transformed vector as we wish to also focus on the robustness to scale. The duplicate-noise transformation, which adds dummy noise dimensions, is motivated by the fact that the number of hidden neurons can change through the layers and often many of these dimensions deeper within the network are usually very sparse and noisy. Similarly, the duplicate-self transformation is another approach to changing the dimensionality of the input while preserving the total information.

**Preprocessing methods:** Czyż et al. (2023)'s work indeed finds that the Gaussianization-based local normalization yields better results than uniform marginalization, when the base distribution is the multivariate student distribution. However, they do note that the improvements are minor, and overall they end up choosing the standard variance-based local normalization over the other two. Our conjecture is that Gaussianization may work better when the distribution has long tails, as long-tail distributions are typically harder to estimate MI for, which was observed in (Czyż et al., 2023). This is potentially because the data samples that are a part of the long tail may end up adding more noise to the final estimate, than in a typical case with Gaussian variables, and Gaussianizing the data preserves MI while avoiding long tails in the input, thereby leading to better performance. A potential direction of future work is thus incorporating similar considerations for our global normalization strategies to further enhance the accuracy of MI estimators.

**Scale invariance:** As the data was already preprocessed using local normalization approaches, scale invariance isn't a part of the analysis in (Czyż et al., 2023), similar to (Kinney & Atwal, 2014). In our case, all our normalization approaches are based on ensuring that the MI estimates are scale-invariant and while retaining other desired properties which may not hold for standard normalization approaches, such as robustness to noisy dimensions.

### 6.3 Neural network-based pre-processing

As one of our contributions is a set of new pre-processing strategies to ensure desirable scale-invariant behaviour of MI estimators, we discuss how this compares to other recent work that uses neural networks to learn pre-processing strategies for more accurate MI Estimation.

Gowri et al. (2024): In this work, the authors propose a two-step MI estimation procedure. First, they train compressed representations of the input which minimize an upper bound on the conditional entropies between the compressed representations and the input/output, after which they estimate the MI between the compressed representations using KSG. In the limiting case, these compressed representations would preserve the true MI, otherwise, normally they are upper-bounded by it. It is worth noting their compressed representations can be of any isomorphic form w.r.t the choice of compressor, and as they use neural networks, the scale of the compression can vary. So to alleviate the issue, they use the standard unit normalization approach (local). As such, our findings in this work, including the different variants proposed for global normalization, can be applied to their KSG estimation step to generate more robust MI estimates.

Butakov et al. (2024): It seemed to us that this work proposed a new estimator using normalizing flows, which in this case are function maps (which can be a neural network) which eventually transform the data distribution to a simpler one which has a closed form solution for MI estimation. In their work, they seem to focus on a Gaussian base distribution. In principle, their estimator could be scale invariant, as their final estimator is the analytical MI expression itself. However, we did not see any explicit theoretical study on this, as one will need to show that the correlation matrix at the end of applying multiple normalizing flows is invariant to one-sided scaling of one of the variables. Overall, our work points out that scale-invariance is an important consideration for any MI estimator to prevent scale confounding in the estimates.

## 7 Conclusion

We presented a comprehensive study of scale invariance in MI estimators, and its impact on estimation accuracy, trends, and on MI-based analysis of neural network training. We outlined multiple normalization approaches to combat scale changes, centered around KSG and MINE, and discussed the pros and cons of each approach. Specifically targeting the high-dimensional and low-data regime, intuitive and empirical

arguments were given for each normalization approach and the final choice of estimators. Overall we found that while both local normalization and global normalization have their own strengths, in most practical scenarios, global normalization variants fare better. Both normalization strategies lead to desirable behaviour in response to input scale changes. Extensive experiments across two broad settings were conducted to measure the overall performance of each estimator. In almost all cases, the local and global normalization approaches fare much better than their unnormalized counterparts, while global normalization variants have the best performance overall. Lastly, on three real datasets, we studied the information plane dynamics w.r.t the hidden layer feature representations during training, for the unnormalized and normalized estimator variants. More clear trends of fitting and compression were observed with global normalization approaches in two out of the three datasets, with KSG-Global variants showing clearer trends than MINE-Global variants. Our work highlights the importance of scale-awareness in the problem of MI estimation, and its potential impact on MI estimates.

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

# A  Appendix: Details for Experiments

## A.1  Details for the Section 5: Experimental Studies

**Key Parameters:**

**Figure 1: Average MI estimates for KSG for a varying number of noise dimensions**

- **Setup:** Additive Gaussian ($X, T \in \mathbb{R}^2$ where $T = X + \epsilon$, with $\epsilon \sim \mathcal{N}\left(0, \sigma^2 I_2\right)$)
- **Number of Samples:** 1000
- **Number of Trials:** 10
- $\sigma = \frac{1}{\sqrt{2}}$
- $\sigma' = 0.04$

**Figure 2: Bias of MINE-based measures for varying noise dimensions**

- **Setup:** Correlated Gaussian $X, T \in \mathbb{R}^2$ with correlated coefficient $\rho$
- **Number of Samples**: 1000
- **Number of Trials:** 10
- **Correlation coefficient**: $\rho$: 0.2

**Figure 3: Bias of the KSG estimator with the real data dimension**

**Figure 3(a):**

- **Setup:** Correlated Gaussian $X, T \in \mathbb{R}^d$ with correlated coefficient $\rho$
- **Number of Samples**: 1000
- **Number of Trials:** 20
- **Dimensionality ($d$):** Evaluated over $d \in \{1, 2, 3, 4, 5, 6, 7, 8, 9\}$

**Figure 3(b):**

- **Setup:** Correlated Gaussian $X, T \in \mathbb{R}^d$ with correlated coefficient $\rho$
- **Number of Samples**: 200
- **Number of Trials:** 20
- **Correlation coefficient**: $\rho \sim Unif[0.4, 0.9]$, where $Unif$ denotes the uniform distribution
- **Dimensionality ($d$):** Evaluated over $d \in \{2, 4, 8, 16, 32, 64\}$, and $\log_2 d \in \{1, 2, 3, 4, 5, 6\}$.

**Figure 4: Data duplication: Average MI estimates for KSG-based approaches.**

- **Setup:** Additive Gaussian ($X, T \in \mathbb{R}^2$ where $T = X + \epsilon$, with $\epsilon \sim \mathcal{N}\left(0, \sigma^2 I_2\right)$), and $X' = [X, X, X, ..X]$.
- **Number of Samples:** 1000
- **Number of Trials:** 10
- $\sigma = \frac{1}{\sqrt{2}}$
- **Dimensionality of Final Input:** $d_{X'} \in \{2, 22, 42, 62, 82, 102, 122, 142, 162, 182\}$

**Figure 5: Estimator bias versus dimension: Comparing MINE with MINE-Global variants**

- **Setup:** Correlated Gaussian $X, T \in \mathbb{R}^d$ with correlated coefficient $\rho$
- **Number of Samples**: 1000
- **Number of Trials:** 10
- **Correlation coefficient**: $\rho$: 0.2
- **Dimensionality ($d$):** Evaluated over $d_X, d_T \in \{1, 2, 3, 4, 5, 6, 7, 8, 9\}$

**Figure 6: Analysis of MI Estimators in response to data scaling. Estimates are for $I(\eta X; T)$, where $\eta$ is the scaling factor**

**Figure 6(a) & 6(b):**

- **Setup:** Correlated Gaussian $X, T \in \mathbb{R}^2$ with correlated coefficient $\rho$, estimates are for $I(\eta X; T)$, where $\eta$ is the scaling factor.
- **Number of Samples:** 1000
- **Number of Trials:** 20
- **Correlation Coefficient:** $\rho$: 0.8 (KSG); 0.5 (MINE)
- **Scaling Factor ($\eta$):**
  - For KSG: $\eta \in [10^{-2}, 10^3]$, equispaced on a $\log_{10}$ scale.
  - For MINE: $\eta \in [10^{-2}, 10]$, equispaced on a $\log_{10}$ scale.

**Figure 6(c) & 6(d):**

- **Setup:** Correlated Gaussian $X, T \in \mathbb{R}^2$ with correlated coefficient $\rho$, RMSE of MI estimates $I(\eta X; T)$, where $\eta$ is the scaling factor.
- **Number of Samples:** 1000
- **Number of Trials:** 20
- **Correlation Coefficient:** $\rho \sim Unif[0, 0.8]$
- **Scaling Factor ($\eta$):**
  - For KSG: $\eta \in [10^{-2}, 10^3]$, equispaced on a $\log_{10}$ scale.
  - For MINE: $\eta \in [10^{-2}, 10]$, equispaced on a $\log_{10}$ scale.
- **Error Metric:** Root Mean Squared Error (RMSE) computed between MI estimates and ground truth.

**Figure 7: Average MI estimates for various estimators across different values of SNR.**

- **Setup:** Additive Gaussian noise base, where $T = X + \epsilon$ with $\epsilon \sim \mathcal{N}(0, \sigma^2 I_2)$. Additionally, $T$ is scaled to $T' = 0.1T$, and the mutual information is estimated as $I(X; T')$.
- **Number of Samples:** 1000
- **Number of Trials:** 10
- $\sigma = \frac{1}{\sqrt{SNR}}$
- **Dimensionality ($d$):** $d_X, d_T = 2$

## B MI Estimators: Configurations

### B.1 KSG

We used the NPEET MI estimator toolbox for estimating KSG and KSG-based measures [3]. We set $k = 3$ for all experiments. For the global KSG variants, we fix $c_1 = 0.1, c_2 = 0.2, ..., c_n = 2$ for all our experiments.

### B.2 MINE

We used the popular pytorch-based package [4] for the MINE implementation.

**Overall MINE implementation:** For estimating $I(X; T)$, we used single-hidden layer relu-activated neural networks of the configuration: $(d_X + d_T) \to H_1 \to H_2 \to ... \to H_k \to 1$, where $d_X + d_T$ is the dimensionality of the input and $H_1, .., H_k$ is the number of hidden neurons for each hidden layer. The last layer is a linear layer. We used the Adam optimizer with a learning rate of 0.001. The hidden neuron configuration varies depending on our experiment. We set the number of epochs to 50 for all experiments. Given a training dataset $S = \{(X_1, T_1), .., (X_n, T_n)\}$, the network $F_\theta$ effectively minimizes the following loss:

$$-\frac{1}{n} \sum_{i=1}^{n} F_\theta\left(X_i, T_i\right) + \log\left(\frac{1}{n} \sum_{i=1}^{n} e^{F_\theta\left(X_i, \tilde{T}_i\right)}\right), \tag{9}$$

Note that we estimated MINE in the standard manner as per its original definition, which is, once the networks are optimized, we estimate the the above loss on the training dataset Song & Ermon (2019). It is noteworthy that to avoid overestimation, (Czyż et al., 2023) proposes an approach where the MINE estimate

---

[3] https://github.com/gregversteeg/NPEET  [4] https://github.com/gtegner/mine-pytorch

is computed on the test data. However, in our case, we find that MINE does not overestimate for most of the cases in Tables 3 and 5, as it almost always has negative bias, and thus the training/test split approach isn't necessary in our case.

**Figures 1-7 and Tables 3 and 5:** For the small dataset cases, we found that using a smaller number of hidden neurons yielded significantly better and more stable results. Thus, for Figures 1-7 and Tables 3 and 5, we found that using a single hidden layer with $H_1 = 20$ yielded the most stable results on average, and thus we set $H_1 = 20$. We found that for this small sample size setting, increasing $H_1$ led to unstable estimates and large variance of estimators.

**Figure 8-10:** For the larger dataset cases, which is the case for our real datasets, we were able to increase $H$, and the details are as follows. For the IB dataset, we have one hidden layer with $H = 30$ neurons. For MNIST and CIFAR-10 datasets, we have two hidden layers with $H = 30$ neurons each. Each dense layer uses ReLU activation.

### B.3 Network Architecture for Neural Network Analysis in Section 6

Table 4: Model Architecture for IB Dataset

| Layer | Dimension | Activation Function |
|---|---|---|
| Input | $28 \times 28$ | - |
| Flatten | 12 | - |
| Dense | 10 | ReLU |
| **Dense** | **7** | **ReLU** |
| Dense | 5 | ReLU |
| Dense | 4 | ReLU |
| Dense | 4 | ReLU |
| Dense | 2 | SoftMax |

Table 5: Model Architecture for MNIST Dataset

| Layer | Dimension | Activation Function |
|---|---|---|
| Input | $28 \times 28$ | - |
| Flatten | 784 | - |
| Dense | 1024 | ReLU |
| **Dense** | **20** | **ReLU** |
| Dense | 20 | ReLU |
| Dense | 20 | ReLU |
| Dense | 10 | SoftMax |

For the MNIST and IB datasets, we replicate the network architectures from Saxe et al. (2018)'s work, using the widely-adopted *ReLU* activation function for the hidden layers. Specifically, for the IB dataset, we utilize a neural network with 7 hidden layers of dimensions 12-10-7-5-4-3-2. For the MNIST dataset, the neural network consists of 6 fully connected layers with dimensions 784-1024-20-20-20-10.

For the CIFAR-10 dataset, we adopt a neural network with 4 convolutional layers, 3 fully connected layers. The tasks for the MNIST and CIFAR-10 datasets involve classifying image inputs into their respective classes, while the task for the IB dataset involves training a binary decision rule based on 12 randomly distributed points. The networks are trained using SGD and cress-entropy loss. We train 2000 epochs for the IB dataset, 200 epochs for the MNIST dataset, and 1000 epochs for the CIFAR-10 dataset.

In Table 5, Table 4 and Table 6, we present the network architecture and output dimensions for each layer of the neural networks used in our study. The layers with bold text are the layers for extracted $Z$.

Table 6: Model Architecture for CIFAR-10 Dataset

| Layer | Dimension | Activation Function |
|---|---|---|
| Input | $32 \times 32 \times 3$ | - |
| Conv2D | $32 \times 32 \times 96$ | ReLU |
| Conv2D | $32 \times 32 \times 96$ | ReLU |
| MaxPooling | $16 \times 16 \times 96$ | - |
| Dropout | (0.5) | - |
| Conv2D | $16 \times 16 \times 192$ | ReLU |
| Conv2D | $16 \times 16 \times 192$ | ReLU |
| **Global AveragePooling** | **192** | **-** |
| Dense | 512 | ReLU |
| Dense | 256 | ReLU |
| Dense | 20 | SoftMax |

For the IB dataset, we trained for 2000 epochs with an SGD optimizer and a learning rate of $5 \times 10^{-3}$. For the MNIST dataset, we trained for 200 epochs with an SGD optimizer and a learning rate of $5 \times 10^{-4}$. For the CIFAR-10 dataset, we trained for 1000 epochs with an SGD optimizer and a learning rate of $1 \times 10^{-3}$. The batch sizes were 256 for the IB dataset, 128 for the MNIST dataset, and 512 for the CIFAR-10 dataset.

## C   Full Results on Synthetic Data

We provide the full results of Tables 2 and 3 of the main paper, in Table 7, and include the full results for the correlated Gaussian distribution base in Table 8. In the following tables, in addition to the normalized root mean squared error, we also report the Spearman correlation and Bias of each estimator, as defined in Section 5.2.1. Table 7 addresses the full KSG and MINE results in the same setting as Tables 2 and 3, and similarly, Table 8 addresses the full KSG and MINE results for the correlated Gaussian distribution base.

These are the specific takeaways from the Spearman correlation and bias results from Tables 7 and 8:

1. Very interestingly, we find that although the prediction error in terms of normalized RMSE quickly becomes higher than random guessing (red entries) for data in higher dimensions, the Spearman correlation of MI estimates with the ground truth MI still stays high in many cases.

2. A particularly notable example of this is for KSG variants, where we find that in spite of normalized RMSE exceeding 1, the KSG variants still have Spearman correlation measures very close to 1, showing that they are very highly rank-correlated with the true MI. It is therefore clear that the large negative bias of KSG variants in high dimensions affects all cases similarly, and thus the dependency between the estimated MI and the true MI remains. This also highlights that using some carefully calibrated ways of estimating MI in high dimensional settings may yield very accurate predictions in terms of normalized RMSE as well.

3. In fact, we find that over all cases in Tables 7 and 8, only the KSG-global variant stays very consistent in terms of high Spearman correlation. Barring only two cases, we see that the Spearman correlation of KSG-global-$L_\infty$ is always greater than 0.9.

4. We find that overall our normalized variants of KSG and MINE showcase bias closest to 0. Interestingly, we see that for all variants, the bias roughly becomes increasingly negative as the dimensionality of the input increases.

Table 7: Comparing performance measures of MI Estimators: Additive Gaussian Noise Base (Full Results)

| Transformation | | | | | d | measure | KSG-Based Measures | | | | | MINE-Based Measures | | | |
|---|---|---|---|---|---|---|---|---|---|---|---|---|---|---|---|
| rn | cb | sg | ds | dn | | | ksg | bi-ksg | ksg-loc | ksg-glo | ksg-glo-$L_\infty$ | mine | mine-loc | mine-glo | mine-glo-corr |
| | ✓ | ✓ | ✓ | | 2 | RMSE-norm | 0.351 | 0.404 | 0.147 | 0.208 | 0.110 | 0.470 | 0.292 | 0.337 | **0.278** |
| | | | | | | spearman | 0.985 | 0.985 | 0.983 | 0.982 | 0.983 | **0.938** | 0.920 | 0.889 | 0.928 |
| | | | | | | bias | -0.191 | -0.226 | -0.072 | -0.105 | -0.047 | -0.246 | -0.130 | -0.161 | -0.117 |
| | | ✓ | ✓ | | 2 | RMSE-norm | 0.286 | 0.335 | 0.060 | 0.084 | 0.050 | 0.445 | 0.255 | 0.275 | **0.233** |
| | | | | | | spearman | 0.992 | 0.992 | 0.988 | 0.989 | 0.990 | **0.938** | **0.929** | 0.917 | **0.932** |
| | | | | | | bias | -0.160 | -0.192 | -0.013 | -0.035 | 0.004 | -0.233 | -0.108 | -0.126 | -0.093 |
| | ✓ | ✓ | | | 2 | RMSE-norm | 0.458 | 0.533 | 0.145 | 0.113 | 0.113 | 1.036 | **0.560** | 0.658 | 0.565 |
| | | | | | | spearman | 0.982 | 0.981 | 0.986 | 0.987 | 0.987 | 0.548 | 0.816 | 0.853 | **0.884** |
| | | | | | | bias | -0.253 | -0.302 | -0.071 | -0.050 | -0.050 | -0.559 | -0.283 | -0.340 | -0.292 |
| | | ✓ | | | 4 | RMSE-norm | 1.275 | 1.424 | 0.312 | 0.300 | 0.301 | 1.302 | 0.720 | 0.968 | 0.684 |
| | | | | | | spearman | 0.988 | 0.991 | 0.993 | 0.995 | 0.995 | 0.805 | **0.930** | 0.901 | **0.922** |
| | | | | | | bias | -1.078 | -1.235 | -0.254 | -0.240 | -0.240 | -1.105 | -0.566 | -0.781 | -0.543 |
| ✓ | | | | ✓ | 4 | RMSE-norm | 0.862 | 0.932 | 0.445 | 0.594 | 0.396 | 0.930 | 0.438 | 0.803 | **0.381** |
| | | | | | | spearman | 0.599 | 0.549 | **0.995** | 0.990 | 0.994 | 0.660 | 0.897 | 0.944 | **0.965** |
| | | | | | | bias | -0.653 | -0.714 | -0.363 | -0.490 | -0.322 | -0.738 | -0.296 | -0.640 | -0.265 |
| | | | ✓ | | 4 | RMSE-norm | 0.332 | 0.342 | **0.304** | 0.520 | **0.297** | 0.276 | **0.369** | 0.642 | 0.375 |
| | | | | | | spearman | 0.994 | 0.994 | 0.995 | 0.994 | 0.994 | 0.931 | 0.922 | 0.922 | **0.951** |
| | | | | | | bias | -0.282 | -0.298 | -0.247 | -0.426 | -0.239 | -0.122 | -0.223 | -0.494 | -0.232 |
| | | | | ✓ | 4 | RMSE-norm | 0.332 | 0.342 | 1.327 | **0.298** | 0.297 | 0.622 | 0.423 | 0.895 | **0.269** |
| | | | | | | spearman | **0.994** | **0.994** | 0.954 | **0.995** | 0.994 | 0.876 | 0.865 | 0.917 | **0.942** |
| | | | | | | bias | -0.282 | -0.298 | -1.107 | -0.239 | -0.239 | -0.486 | **0.002** | -0.719 | 0.118 |
| ✓ | ✓ | | | | 4 | RMSE-norm | 1.131 | 1.233 | 0.977 | 0.959 | **0.931** | 1.574 | 1.099 | 1.219 | 1.162 |
| | | | | | | spearman | 0.588 | 0.534 | **0.975** | 0.961 | 0.969 | 0.542 | **0.905** | 0.837 | 0.769 |
| | | | | | | bias | -0.942 | -1.045 | -0.816 | -0.796 | -0.773 | -1.198 | -0.914 | -1.012 | -0.960 |
| | | ✓ | | ✓ | 4 | RMSE-norm | 1.275 | 1.424 | 1.334 | 0.300 | 0.301 | 1.335 | 0.423 | 0.895 | **0.286** |
| | | | | | | spearman | **0.988** | **0.991** | 0.916 | **0.995** | 0.995 | 0.807 | 0.870 | 0.914 | **0.939** |
| | | | | | | bias | -1.078 | -1.235 | -1.113 | -0.239 | -0.240 | -1.119 | -0.004 | -0.718 | 0.148 |
| ✓ | ✓ | ✓ | | ✓ | 6 | RMSE-norm | 1.981 | 2.129 | 1.904 | 1.021 | 1.021 | 1.881 | 0.570 | 1.516 | **0.437** |
| | | | | | | spearman | 0.969 | 0.966 | 0.905 | 0.989 | 0.989 | 0.852 | 0.786 | **0.881** | 0.855 |
| | | | | | | bias | -1.999 | -2.174 | -1.919 | -1.021 | -1.021 | -1.898 | -0.179 | -1.502 | -0.181 |
| | | ✓ | | | 6 | RMSE-norm | 1.983 | 2.131 | **0.816** | 0.811 | 0.812 | 1.843 | 1.147 | 1.535 | 1.185 |
| | | | | | | spearman | 0.959 | 0.957 | **0.995** | 0.995 | 0.995 | 0.842 | 0.771 | **0.862** | 0.815 |
| | | | | | | bias | -2.001 | -2.175 | **-0.816** | -0.809 | -0.810 | -1.861 | -1.116 | -1.527 | -1.155 |
| ✓ | | | ✓ | | 6 | RMSE-norm | 1.377 | 1.408 | 1.343 | 1.605 | **1.290** | 1.213 | **0.991** | 1.444 | 1.004 |
| | | | | | | spearman | **0.983** | **0.981** | 0.979 | 0.968 | **0.989** | 0.694 | 0.783 | 0.831 | **0.881** |
| | | | | | | bias | -1.377 | -1.410 | -1.347 | -1.609 | -1.291 | -1.158 | -0.933 | -1.432 | -0.971 |
| ✓ | | | ✓ | | 6 | RMSE-norm | 1.643 | 1.730 | 1.275 | 1.660 | 1.153 | 1.014 | 1.088 | 1.441 | 1.058 |
| | | | | | | spearman | 0.338 | 0.314 | 0.937 | 0.871 | **0.976** | 0.775 | 0.798 | **0.841** | 0.818 |
| | | | | | | bias | -1.631 | -1.730 | -1.275 | -1.666 | -1.150 | -0.967 | -1.042 | -1.431 | -1.028 |
| | | ✓ | | ✓ | 6 | RMSE-norm | 1.983 | 2.131 | 1.905 | 0.814 | 0.816 | 1.831 | **0.517** | 1.499 | 0.545 |
| | | | | | | spearman | 0.956 | 0.955 | 0.926 | 0.995 | 0.995 | 0.813 | 0.806 | 0.811 | **0.817** |
| | | | | | | bias | -2.001 | -2.175 | -1.921 | -0.811 | -0.814 | -1.847 | -0.068 | -1.471 | 0.391 |

Table 8: Comparing performance measures of MI Estimators: Correlated Gaussian Base

| Transformation | | | | | N | d | measure | KSG-Based Measures | | | | | MINE-Based Measures | | | |
| rm | cb | sg | ds | dn | | | | ksg | bi-ksg | ksg-loc | ksg-glo | ksg-glo-$L_\infty$ | mine | mine-loc | mine-glo | mine-glo-corr |
|---|---|---|---|---|---|---|---|---|---|---|---|---|---|---|---|---|
| | | | | | 200 | 2 | RMSE-norm | 0.125 | 0.125 | 0.122 | 0.113 | 0.115 | 0.573 | 0.579 | 0.601 | 0.555 |
| | | | | | | | spearman | 0.918 | 0.922 | 0.924 | 0.938 | 0.936 | 0.478 | 0.711 | 0.596 | 0.801 |
| | | | | | | | bias | -0.021 | -0.023 | -0.020 | 0.014 | 0.015 | -0.238 | -0.247 | -0.257 | -0.238 |
| | | | ✓ | | 200 | 2 | RMSE-norm | 0.138 | 0.140 | 0.140 | 0.160 | 0.116 | 0.359 | 0.321 | 0.475 | 0.315 |
| | | | | | | | spearman | 0.923 | 0.923 | 0.914 | 0.898 | 0.878 | 0.886 | 0.929 | 0.931 | 0.940 |
| | | | | | | | bias | -0.049 | -0.051 | -0.048 | -0.042 | -0.005 | -0.114 | -0.089 | -0.192 | -0.108 |
| | ✓ | | ✓ | | 200 | 2 | RMSE-norm | 0.305 | 0.332 | 0.281 | 0.324 | 0.233 | 0.850 | 0.416 | 0.544 | 0.405 |
| | | | | | | | spearman | 0.794 | 0.798 | 0.911 | 0.916 | 0.867 | 0.586 | 0.889 | 0.907 | 0.944 |
| | | | | | | | bias | -0.124 | -0.146 | -0.117 | -0.125 | -0.077 | -0.282 | -0.149 | -0.227 | -0.156 |
| | ✓ | ✓ | ✓ | | 1000 | 2 | RMSE-norm | 0.129 | 0.151 | 0.072 | 0.094 | 0.060 | 0.278 | 0.135 | 0.195 | 0.155 |
| | | | | | | | spearman | 0.961 | 0.958 | 0.941 | 0.960 | 0.952 | 0.951 | 0.942 | 0.969 | 0.967 |
| | | | | | | | bias | -0.046 | -0.064 | -0.019 | -0.023 | 0.000 | -0.106 | -0.029 | -0.057 | -0.038 |
| | | ✓ | ✓ | | 1000 | 2 | RMSE-norm | 0.090 | 0.103 | 0.046 | 0.052 | 0.049 | 0.274 | 0.130 | 0.169 | 0.144 |
| | | | | | | | spearman | 0.947 | 0.950 | 0.954 | 0.943 | 0.961 | 0.957 | 0.941 | 0.973 | 0.969 |
| | | | | | | | bias | -0.031 | -0.041 | -0.004 | -0.003 | 0.017 | -0.104 | -0.025 | -0.042 | -0.032 |
| | ✓ | ✓ | | | 1000 | 2 | RMSE-norm | 0.126 | 0.147 | 0.065 | 0.060 | 0.060 | 0.489 | 0.281 | 0.358 | 0.290 |
| | | | | | | | spearman | 0.940 | 0.935 | 0.939 | 0.949 | 0.950 | 0.879 | 0.950 | 0.946 | 0.950 |
| | | | | | | | bias | -0.042 | -0.060 | -0.020 | 0.000 | 0.000 | -0.213 | -0.105 | -0.145 | -0.112 |
| | | ✓ | | | 200 | 5 | RMSE-norm | 0.962 | 1.088 | 0.481 | 0.470 | 0.469 | 1.014 | 0.949 | 0.982 | 0.934 |
| | | | | | | | spearman | 0.736 | 0.408 | 0.964 | 0.966 | 0.969 | 0.715 | 0.792 | 0.852 | 0.789 |
| | | | | | | | bias | -0.664 | -0.822 | -0.322 | -0.293 | -0.294 | -0.703 | -0.641 | -0.673 | -0.629 |
| ✓ | | | | | 200 | 5 | RMSE-norm | 0.738 | 0.783 | 0.688 | 0.628 | 0.625 | 0.974 | 0.956 | 0.993 | 0.952 |
| | | | | | | | spearman | 0.713 | 0.569 | 0.893 | 0.895 | 0.905 | 0.606 | 0.804 | 0.765 | 0.833 |
| | | | | | | | bias | -0.442 | -0.498 | -0.406 | -0.349 | -0.346 | -0.672 | -0.648 | -0.681 | -0.649 |
| | ✓ | | | | 200 | 5 | RMSE-norm | 0.778 | 0.819 | 0.726 | 0.695 | 0.693 | 1.084 | 0.980 | 0.999 | 0.967 |
| | | | | | | | spearman | 0.933 | 0.931 | 0.932 | 0.937 | 0.939 | 0.605 | 0.773 | 0.805 | 0.711 |
| | | | | | | | bias | -0.539 | -0.589 | -0.498 | -0.462 | -0.460 | -0.784 | -0.670 | -0.688 | -0.660 |
| | | | | ✓ | 1000 | 5 | RMSE-norm | 0.258 | 0.259 | 0.793 | 0.253 | 0.253 | 0.514 | 0.489 | 0.680 | 0.285 |
| | | | | | | | spearman | 0.990 | 0.989 | 0.812 | 0.985 | 0.986 | 0.974 | 0.847 | 0.968 | 0.954 |
| | | | | | | | bias | -0.149 | -0.152 | -0.486 | -0.134 | -0.134 | -0.304 | 0.171 | -0.441 | 0.076 |
| ✓ | ✓ | | | | 1000 | 5 | RMSE-norm | 0.898 | 0.981 | 0.792 | 0.767 | 0.744 | 0.952 | 0.767 | 0.819 | 0.766 |
| | | | | | | | spearman | 0.582 | 0.433 | 0.943 | 0.963 | 0.965 | 0.511 | 0.923 | 0.924 | 0.902 |
| | | | | | | | bias | -0.523 | -0.615 | -0.472 | -0.456 | -0.440 | -0.680 | -0.526 | -0.567 | -0.533 |
| | ✓ | | | | 200 | 10 | RMSE-norm | 1.381 | 1.473 | 1.008 | 0.999 | 1.000 | 1.441 | 1.357 | 1.424 | 1.352 |
| | | | | | | | spearman | 0.131 | 0.170 | 0.956 | 0.962 | 0.963 | 0.512 | 0.790 | 0.806 | 0.856 |
| | | | | | | | bias | -1.351 | -1.522 | -0.985 | -0.957 | -0.960 | -1.412 | -1.288 | -1.388 | -1.290 |
| ✓ | ✓ | ✓ | | ✓ | 200 | 10 | RMSE-norm | 1.424 | 1.514 | 1.384 | 1.122 | 1.121 | 1.446 | 1.063 | 1.411 | 1.096 |
| | | | | | | | spearman | -0.127 | -0.083 | 0.751 | 0.935 | 0.941 | 0.590 | 0.005 | 0.771 | 0.786 |
| | | | | | | | bias | -1.212 | -1.383 | -1.175 | -0.910 | -0.909 | -1.418 | -0.451 | -1.362 | -0.784 |
| | ✓ | | ✓ | | 200 | 10 | RMSE-norm | 1.356 | 1.412 | 1.229 | 1.321 | 1.213 | 1.287 | 1.163 | 1.382 | 1.162 |
| | | | | | | | spearman | 0.831 | 0.823 | 0.872 | 0.885 | 0.900 | 0.652 | 0.933 | 0.942 | 0.916 |
| | | | | | | | bias | -1.164 | -1.263 | -1.043 | -1.111 | -1.011 | -1.182 | -0.990 | -1.325 | -0.976 |
| ✓ | | | ✓ | | 1000 | 10 | RMSE-norm | 1.217 | 1.260 | 1.158 | 1.353 | 1.063 | 0.937 | 0.886 | 1.077 | 0.912 |
| | | | | | | | spearman | 0.815 | 0.641 | 0.977 | 0.947 | 0.986 | 0.956 | 0.954 | 0.982 | 0.969 |
| | | | | | | | bias | -1.033 | -1.120 | -0.976 | -1.145 | -0.890 | -0.869 | -0.844 | -1.030 | -0.861 |
| ✓ | | ✓ | | ✓ | 1000 | 10 | RMSE-norm | 1.424 | 1.516 | 1.386 | 0.875 | 0.877 | 1.185 | 0.808 | 1.090 | 0.723 |
| | | | | | | | spearman | -0.065 | -0.450 | 0.845 | 0.992 | 0.992 | 0.748 | 0.875 | 0.951 | 0.898 |
| | | | | | | | bias | -1.212 | -1.386 | -1.177 | -0.721 | -0.723 | -1.145 | 0.444 | -0.981 | 0.560 |
| ✓ | | | | ✓ | 1000 | 2 | RMSE-norm | 0.141 | 0.159 | 0.093 | 0.078 | 0.061 | 0.280 | 0.157 | 0.235 | 0.150 |
| | | | | | | | spearman | 0.968 | 0.943 | 0.967 | 0.964 | 0.967 | 0.793 | 0.953 | 0.960 | 0.950 |
| | | | | | | | bias | -0.040 | -0.050 | -0.031 | -0.014 | 0.000 | -0.105 | -0.017 | -0.085 | -0.027 |

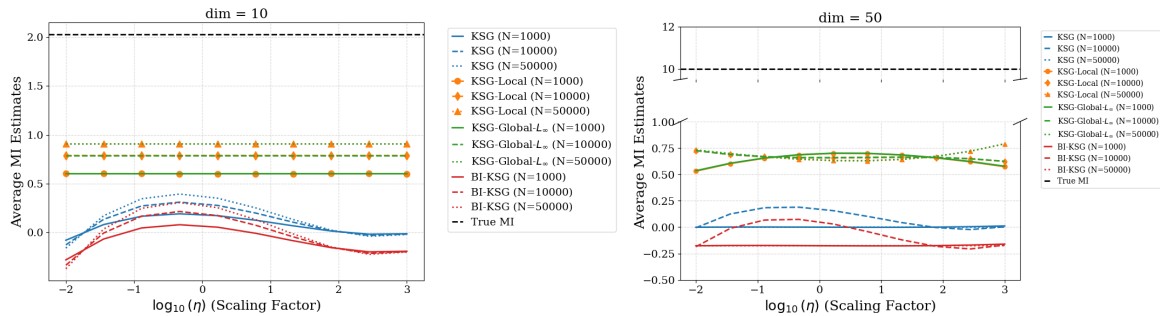

Figure 10: Analysis of the KSG Estimators in response to data scaling for varying number of data samples $N = \{1000, 10000, 50000\}$, and different data dimensionality $d = \{10, 50\}$. Estimates are for $I(\eta X; T)$, where $\eta$ is the scaling factor.

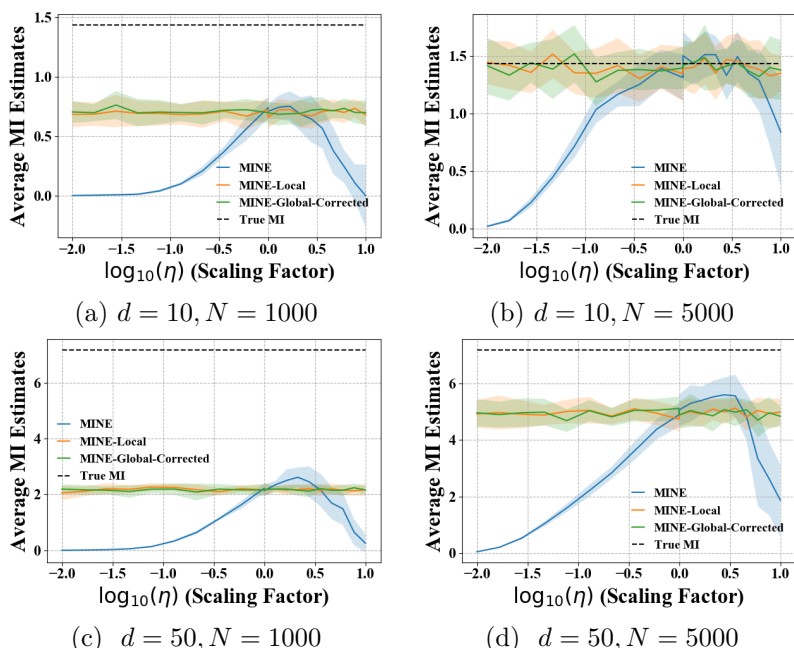

(a) $d = 10, N = 1000$    (b) $d = 10, N = 5000$

(c) $d = 50, N = 1000$    (d) $d = 50, N = 5000$

Figure 11: Analysis of the MINE Estimators in response to data scaling for varying number of data samples $N = \{1000, 5000\}$, and different data dimensionality $d = \{10, 50\}$. Estimates are for $I(\eta X; T)$, where $\eta$ is the scaling factor.

## D    Additional Results

### D.1    Scale Invariance Testing

In the same setting as Section 5.1.4, we conduct more experiments to see if behaviour of the various estimators in response to data scaling remains unchanged for different number of sampled datapoints $N = \{1000, 5000\}$ and $d = \{10, 50\}$. The results are shown in Figures 10 and 11.

**KSG:** Overall, for KSG (Figure 10), when we're analyzing the behaviour of the native estimators, we find that they show similar response to scale, i.e., they converge to very low values near zero as the scale $\eta$ goes to either extremity. In contrast, the scale-invariant estimators preserve the response across scales for $d = 10$. For higher dimensionality $d = 50$, we see some interesting behavorial changes for the scale-invariant KSG-Local and KSG-Global variants. We find that although they don't reach negligible values when scale reaches either extreme, the average MI estimates do not stay the same for all $\eta$, and there is some variation. Overall, the KSG-Global and Local variants behave similarly as before, and their average measures are relatively

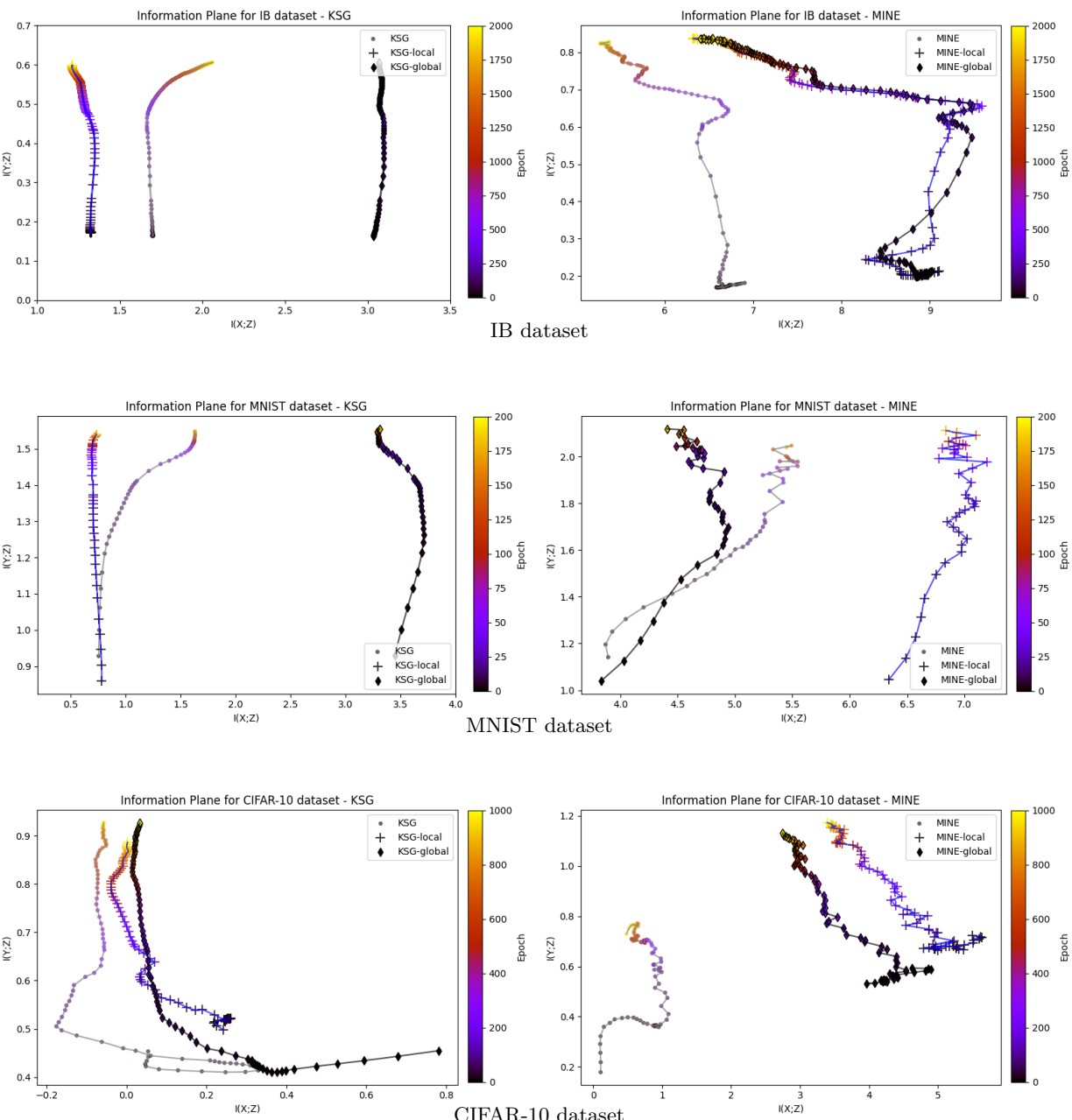

Figure 12: Information plane ($I(X; Z)$ against $I(Y; Z)$) for IB, MNIST and CIFAR-10 datasets. $Z$ is the output of $3^{rd}$ layer for IB dataset and MNIST dataset, and $7^{th}$ layer for CIFAR-10 dataset. Details in Appendix B.3

stable when $\eta$ is no not small or too large. Only when $\eta$ is either less than 0.1 or greater than 10, we see a slightly more pronounced change for the average MI estimates.

**MINE:** For the MINE results (Figure 11), we observe some interesting variations in the scale response depending on the dimensionality and the number of datapoints. We find that when we use a greater number of datapoints, then the MINE estimate converges to zero only for smaller scaling factors $\eta$. Our result in Proposition 5 essentially applies for the limiting case when $\eta \to 0^+$, and we can intuitively show that the value of $\eta$ for which we see this limiting behaviour reduces with greater sample number. The proof of Proposition 5 essentially finds that the weights associated with the scaled RV $\eta T$ is upper bounded in magnitude by a multiplicative factor of $\eta$ and the number of updates and epochs. As a greater number of

datapoints also implies a greater number of gradient descent updates, this also implies that we shall have potentially larger weights when the sample number increases. This implies that for larger datapoints $N$ the MI estimate may stay non-zero for a larger range of $\eta$, and drop to near-zero values only near the extreme values of $\eta$, which is what we see in Figure 11. Lastly, we see that the local and global variants of MINE have relatively stable behaviour of their mean values across $\eta$ for all $d, N$ combinations. However, we do see that the variance of these estimators increases with more datapoints.

## E    Information Plane Analysis

In figure 12, we plot the information plane for the IB, MNIST, and CIFAR-10 datasets using the KSG and MINE estimators to reveal MI changes in neural network training. The displayed results represent the averages from 10 trials. For IB and MNIST datasets, both $I(X; Z)$ and $I(Y; Z)$ obtained by the original KSG estimator generally increase with the number of training epochs. In contrast, the KSG-local and KSG-global-$L_\infty$ estimators demonstrate a more refined information bottleneck trend. Specifically, these estimators show a clear fitting phase where $I(X; Z)$ initially increases and then stabilizes, followed by a compression phase where $I(X; Z)$ decreases while $I(Y; Z)$ remains monotonically increasing. Among these results, the KSG-global estimator yields the most consistent trends, exhibiting the most distinct fitting and compression phases in two of the three datasets, thus effectively capturing the information bottleneck phenomenon.

The information plane plots using the MINE estimator for the IB, MNIST, and CIFAR-10 datasets reveal different levels of performance. The original MINE estimator fails to effectively capture the information bottleneck phenomenon in all three datasets. However, the MINE-global-corrected variant demonstrates an ability to observe the information bottleneck trend in all three datasets. The fitting and compression trends can also be observed for MINE-local, but the compression trends are harder to decipher clearly on MNIST and CIFAR-10. In general, we find that the MINE estimates are significantly noisier than the KSG estimates.

## F    Proofs of theoretical results

**Proposition 1.** It holds that $\widehat{I}^n_{bin}(\alpha X; \alpha T) = \widehat{I}^n_{bin}(X; T)$ and $\widehat{I}^n_{bin}(X; \alpha T) = \widehat{I}^n_{bin}(X; T) \ \forall \alpha \in \mathbb{R}^+$.

*Proof.* We note that the number of bins chosen for each dimension is fixed, and the locations of the bins are determined by the minimum and maximum values of the data in each dimension, i.e., they determine the edges of the bins. Let $X_{min} \in \mathbb{R}^d$ then denote the vector of minimum values across all dimensions, and vice-versa for $X_{max}$. When $X$ scales to $\alpha X$, as $\alpha > 0$, we have that the vector of minimum values for $\alpha X$ is simply $\alpha X_{min}$ and similarly for $X_{max}$, and the binning locations also get scaled by $\alpha$. Thus, there is a bijection between the binning locations of $X$ and $\alpha X$. Since both the binning structure and the data points within each bin are scaled uniformly, the probability of data falling into any given bin remains unchanged. Therefore, the distribution of data across the bins is invariant under scaling, leading to the same binning estimator $\widehat{I}^n_{bin}(X; T) = \widehat{I}^n_{bin}(\alpha X; T) = \widehat{I}^n_{bin}(\alpha X; \alpha T)$ for any scaling factor $\alpha$. $\qquad\square$

**Proposition 2.** It holds that $\widehat{I}^n_{KSG}(\alpha X; \alpha T) = \widehat{I}^n_{KSG}(X; T)$, $\forall \alpha \in \mathbb{R}^+$.

*Proof.* We note the expression for the KSG estimator (equation 3 from (Kraskov et al., 2004) as follows:

$$\widehat{I}^n_{KSG}(\alpha X; \alpha T) = \psi(k) + \psi(n) - \frac{1}{k} - \frac{1}{n} \sum_{i=1}^{n} \left( \psi(n_{\alpha x, i, \infty}) + \psi(n_{\alpha t, i, \infty}) \right) \tag{10}$$

Here, $\psi$ denotes the digamma function(Abramowitz, 1974), and $n_{\alpha x, i, \infty} = \sum_{j \neq i} \mathbb{I}\{||\alpha X_i - \alpha X_j||_\infty \leq \rho_{k, i, \infty}\}$, where $\rho_{k, i, \infty}$ is the k-NN distance of the joint sample $i$, $\{\alpha X, \alpha T\}$ (this distance is computed in $d + m$ dimensions). Furthermore, $||\alpha X_i - \alpha X_j||_\infty$ represents the $X$-dimensions only distance (i.e. in $d$ dimensional space). Let $\rho'_{k, i, \infty}$ be the k-NN distance of the joint sample $i$ for the unscaled variables $\{X, T\}$. It is trivial to see that $\rho_{k, i, \infty} = \alpha \rho'_{k, i, \infty}$ . Thus, $n_{\alpha x, i, \infty} = \sum_{j \neq i} \mathbb{I}\{||\alpha X_i - \alpha X_j||_\infty \leq \alpha \rho'_{k, i, \infty}\} = \sum_{j \neq i} \mathbb{I}\{||X_i - X_j|| \leq \rho'_{k, i, \infty}\} = n_{x, i, \infty}$, and similarly $n_{\alpha t, i, \infty} = n_{t, i, \infty}$. This shows that $\widehat{I}^n_{KSG}(\alpha X; \alpha T) = \widehat{I}^n_{KSG}(X; T)$. $\qquad\square$

**Proposition 3.** It holds that $\lim_{\alpha,n\to\infty} \widehat{I}^n_{KSG}(X;\alpha T) = -\frac{1}{k}$ and $\lim_{\alpha\to 0^+,n\to\infty} \widehat{I}^n_{KSG}(X;\alpha T) = -\frac{1}{k}$, where $k$ is the k-nearest neighbor parameter for the estimator. Thus, $\widehat{I}^n_{KSG}(X;\alpha T)$ need not be equal to $\widehat{I}^n_{KSG}(X;T)$.

*Proof.* Following from the proof of Proposition 2, we note that as $\alpha\to 0$, we first show $n_{\alpha t,i,\infty} = \sum_{j\neq i} \mathbb{I}\{||\alpha T_i - \alpha T_j||_\infty \leq \rho_{k,i,\infty}\}\to n$. First, distances in the joint space $(X,\alpha T)$ can be expressed as $||(X_i,\alpha T_i) - (X_j,\alpha T_j)||_\infty$. Thus, as $\alpha\to 0$, $\rho_{k,i,\infty}$ becomes the k-nearest neighbor distance in the $X$-space only. Next, because $X$ and $T$ are bounded, and when $\alpha\to 0$, the distance $||\alpha T_i - \alpha T_j||_\infty \to 0$. Thus, $n_{\alpha t,i,\infty} = \sum_{j\neq i} \mathbb{I}\{||\alpha T_i - \alpha T_j||_\infty \leq \rho_{k,i,\infty}\}\to n$, as all points are essentially at a zero $T$-only distance between each other in $T$-space. Similarly, $n_{x,i,\infty} = k$ in this case, as the nearest neighbor distance in the joint $(X,\alpha T)$ space becomes $X$-only distance, and $n_{x,i,\infty} = \sum_{j\neq i} \mathbb{I}\{||X_i - X_j||_\infty \leq \rho_{k,i,\infty}\} = k$.

Recalling the expression for KSG, we have:

$$\widehat{I}^n_{KSG}(X;\alpha T) = \psi(k) + \psi(n) - \frac{1}{k} - \frac{1}{n}\sum_{i=1}^{n}\left(\psi(n_{x,i,\infty}) + \psi(n_{\alpha t,i,\infty})\right) \tag{11}$$

Thus we then have: $\lim_{\alpha,n\to\infty} \widehat{I}^n_{KSG}(X;\alpha T) = \psi(k) + \psi(n) - \frac{1}{k} + \frac{1}{n}\sum_{i=1}^{n}\left(\psi(k) + \psi(n)\right)) = -\frac{1}{k}$.

Lastly, as KSG is global scale-invariant (Proposition 2), we have that $\lim_{\alpha\to 0,n\to\infty} \widehat{I}^n_{KSG}(X;\alpha T) = \lim_{\alpha\to 0,n\to\infty} \widehat{I}^n_{KSG}(\frac{1}{\alpha}X;T) = \lim_{\alpha,n\to\infty} \widehat{I}^n_{KSG}(\alpha X;T) = -\frac{1}{k}$. The final result follows from the fact that $\widehat{I}^n_{KSG}(X;T) = \widehat{I}^n_{KSG}(T;X)$. $\square$

**Proposition 4.** It holds that $\widehat{I}^n_{MINE-opt}(X;\alpha T) = \widehat{I}^n_{MINE-opt}(X;T) \ \forall\alpha \in \mathbb{R}^+$.

*Proof.* To demonstrate that $\widehat{I}^n_{MINE-opt}(X;\alpha T) = \widehat{I}^n_{MINE-opt}(X;T)$, we begin by considering any neural network function $f$ that yields a specific value for the expression $\mathbb{E}_{X,Y\sim P(X,T)}[f(X,T)] - \mathbb{E}_{X,T\sim P(X)\times P(T)}\left[e^{f(X,T)}\right]$, there exists a corresponding neural network function $f'$ such that $\mathbb{E}_{X,\alpha T\sim P(X,\alpha T)}[f'(X,\alpha T)] - \mathbb{E}_{X,\alpha T\sim P(X)\times P(\alpha T)}\left[e^{f'(X,\alpha T)}\right]$ has the same value of the expression involving $f$ and vice-versa. To construct $f'$, let $W_T$ be the weights of the first layer of the network $f$ that are attached to $T$, and similarly $W'_T$ for $f'$. Define a new network function $f'$ with the same architecture as $f$ except that $W'_T = W_T/\alpha$. By construction, the function $f'$ satisfies $f'(X,\alpha T) = f(X,T)$, which implies that for every function $f$ that optimizes the expression in equation 6 there is a corresponding function $f'$ for the variables $X$ and $\alpha T$. This also shows that the optimization for $I(X;T)$ and $I(X;\alpha T)$ as expressed in equation 6 is equivalent. As a result, the mutual information estimator $\widehat{I}^n_{MINE-opt}$, which corresponds to the supremum of the value of this expression over all possible neural network functions, is invariant under scaling of $T$. Therefore, we conclude that $\widehat{I}^n_{MINE-opt}(X;\alpha T) = \widehat{I}^n_{MINE-opt}(X;T)$.

$\square$

**Proposition 5.** Consider the MINE optimization problem with input data $S = \{(\alpha X_1, Y_1),...,(\alpha X_n, Y_n)\}$ where $X \in \mathbb{R}^{d_x}$, $Y \in \mathbb{R}^{d_y}$, $(X,Y) \sim P(X,Y)$ are bounded RVs and $\alpha \in \mathbb{R}^+$ is a scaling factor. We consider a neural network of depth $d_n + 1$ having $h_1, h_2, .., h_{d_n}$ relu-activated hidden neurons in the respective layers. The network is trained via gradient descent on the MINE loss function in equation 7 for a finite number of epochs $n_e$. Let the *trained* weights between the $j^{th}$ node of the $l + 1^{th}$ hidden layer and the $i^{th}$ node of the $l^{th}$ hidden layer be denoted by $w^l_{ji} \in \mathbb{R}^d$. We consider the case where the initialized weights are very close to zero but not exactly zero (to allow unsymmetrical learning). We assume that the network weights are bounded, such that every weight $|w^l_{ji}| \leq B$ for some $B \in \mathbb{R}$. Let $\eta(t)$ denote the learning rate used at epoch $t$. Then we have, $\forall i,j$,

$$\lim_{\alpha\to 0^+} \left|w^1_{ji}\right| = 0 \tag{12}$$

*Proof.* Let $X$ be represented in terms of its individual dimension RVs as $X = [x_1, x_2, .., x_{d_x}]$. The weight update rule for $w^1_{ji}$, at epoch $t$ is then

$$\Delta w^1_{ji} = -\eta(t)\alpha x_i \delta^1_j(t). \tag{13}$$

Here $\delta_j^1(t)$ is the backprop error signal at the $j^{th}$ node of the second layer. Let $a(.)$ denote the relu activation function, and $a'$ its derivative. We then note the following chain rule for the error signal:

$$\delta_j^{l-1}(t) = \sum_{i=1}^{h_l} \delta_j^l(t) w_{ji}^{l-1} a'(z_j^{l-1}(t)), \tag{14}$$

where $z_j^{l-1}(t)$ denotes the output of the $j^{th}$ node of the $l-1^{th}$ layer itself. As the weights are bounded, and the relu derivative is bounded by 1, we can write:

$$\delta_j^{l-1}(t) \leq \sum_{i=1}^{h_l} \delta_j^l(t) B, \tag{15}$$

Let us assume the whole network function can be denoted as $f_W(X, Y) : \mathbb{R}^{d_x + d_y} \to \mathbb{R}$. Note that the network outputs a single real number, and the last layer does not have any activation function (which is relu for the other layers). In the context of MINE's optimization problem, the network minimizes the following loss function:

$$\widehat{I}_{\mathrm{MINE}}(X; Y) = -\frac{1}{n} \sum_{i=1}^n f_W(\alpha X_i, Y_i) + \log \left( \frac{1}{n} \sum_{i=1}^n e^{f_W(\alpha X_i, \tilde{Y}_i)} \right), \tag{16}$$

The error signal at the last layer, $\delta^{d_n+1}(t)$ is the derivative of the loss w.r.t the network output. However, as the MINE optimization effectively has two distributions $P(X, Y)$ and $P(X)P(Y)$ of input, we consider the error signal pertaining to these distributions separately. The loss function for a datapoint $X_j, Y_j \sim P(X, Y)$ is $-\frac{1}{n} f_W(\alpha X_j, Y_j)$, which yields an error signal $\delta^{d_n+1}(t) = -1/n$. The loss function for a datapoint $X_j, \tilde{Y}_j \sim P(X)P(Y)$ is $\log \left( \frac{1}{n} \sum_{i=1}^n e^{f_W(\alpha X_i, \tilde{Y}_i)} \right)$, which yields an error signal

$$\delta^{d_n+1}(t) = \frac{d \left( \log \left( \frac{1}{n} \sum_{i=1}^n e^{f_W(\alpha X_i, \tilde{Y}_i)} \right) \right)}{d f_W(\alpha X_j, \tilde{Y}_j)} = \frac{e^{f_W(\alpha X_j, \tilde{Y}_j)}}{\sum_{i=1}^n e^{f_W(\alpha X_i, \tilde{Y}_i)}} \leq 1 \tag{17}$$

Thus, across both cases, we have that the error signal $\left| \delta^{d_n+1}(t) \right| \leq 1$.

This observation coupled with applying the result in equation 15 yields:

$$\left| \delta_j^{l-1}(t) \right| \leq B^{d_n} \prod_{i=2}^{d_n} h_i \tag{18}$$

Next, as $X$ and $Y$ are bounded, we can assume that every dimension of $|x_i| \leq K$ for some $K \in \mathbb{R}^+$. Let us set $C = K B^{d_n} \prod_{i=2}^{d_n} h_i$. With this, we have that $|\Delta w_{ji}^1| \leq \eta(t) \alpha C$. Let the number of mini-batch updates per epoch be $b_u$. With this, the total change in $|\Delta w_{ji}^1|$ can be bounded as:

$$|\Delta_{total} w_{ji}^1| \leq \alpha \left( \sum_{t=1}^{n_e} \eta(t) b_u C \right) \tag{19}$$

The trained weight $|w_{ji}^1| \leq \epsilon + \alpha \left( \sum_{t=1}^{n_e} \eta(t) b_u C \right)$ where $\epsilon \to 0$ is the intialized value of the corresponding weight. Thus, when $\alpha \to 0^+$, we have that $\epsilon + \alpha \left( \sum_{t=1}^{n_e} \eta(t) b_u C \right) \to 0$, yielding $\lim_{\alpha \to 0^+} \left| w_{ji}^1 \right| = 0$. $\qquad \square$

**Proposition 6.** We consider the same setting as Proposition 5 for the MINE estimation problem. There, it holds that $\lim_{\alpha \to 0} \widehat{I}_{MINE-sgd}^n(X; \alpha T) = 0$ . Thus, $\widehat{I}_{MINE-sgd}^n(X; \alpha T)$ need not be equal to $\widehat{I}_{MINE-sgd}^n(X; T)$.

*Proof.* Let $f^*$ represent the neural network function which optimizes the expression for $\widehat{I}^n_{MINE-sgd}(X;\alpha T)$, which is $\mathbb{E}_{X,\alpha T \sim P(X,\alpha T)}\left[f'(X,\alpha T)\right] - \mathbb{E}_{X,\alpha T \sim P(X) \times P(\alpha T)}\left[e^{f'(X,\alpha T)}\right]$, via SGD. Let $W_T$ be the weights of the first layer of the network $f^*$ which are attached to $\alpha T$. From Proposition 5, we have that $\lim_{\alpha \to 0^+}\left|w^1_{ji}\right| = 0$. Thus, as $\alpha \to 0$, the weights $W_T \to 0$ as well. This indicates that the contribution of $T$ to the function $f^*$, as $\alpha \to 0$, will be negligible. Thus effectively, $\lim_{\alpha \to 0}\widehat{I}^n_{MINE-sgd}(X;\alpha T) = \widehat{I}^n_{MINE-sgd}(X;0) = 0$. This holds mainly because when $W_T \to 0$, the network essentially ignores the changes in $\alpha T$, and thus $\alpha T$ is essentially treated as a constant variable. This proves the result. $\qquad\square$

