# OpenReview forum: "Towards Robust Scale-Invariant Mutual Information Estimators"
_TMLR — Rejected by TMLR_

### Review · Reviewer_f2tg · 2024-10-09

**Summary Of Contributions:**

This paper investigates the effect of scaling of random variables on mutual information estimators.
The paper presents three estimators (binning, KSG and MINE) and :
- gives theoretical results (with proof sketches) on whether or not these estimators are scale invariant.
- proposes normalization strategies to remedy the problem of scale sensitity of KSG and MINE.
- investigates numerically on simulated and benchmark datasets the effect of scaling on different versions of KSG and MINE.
- gives recommendations on which scale invariant version KSG and MINE to use.

**Audience:**

Yes

**Broader Impact Concerns:**

Not concerned.

**Claims And Evidence:**

Yes

**Requested Changes:**

Adjustments that are critical to secure my recommendation for acceptance:

1) In view of Weaknesses 1 and 2, define each term such that the paper is self-contained, and give more details on the proofs, especially the proof of Proposition 5.

2) In view of Weakness 3°, I suggest to add confidence intervals on the plots.

Proposed adjustments to strengthen the work:

A) Organization : Section 6 is devoted to "experimental studies" ; but some experimental results are detailed in the previous section. The organization could be made such that it is easier to follow. For instance, the authors mention in Section 5 they use correlated Gaussians, but the precise definition of correlated Gaussian is given in Section 6.

B) The three definitions page 7 are not written rigorously. For instance, RVs $X_1,\dots,X_n$ are introduced but are not used. It is strange to integrate "over $S$" the RV $X$.

Minor remarks:

(i) Below Eq. (7), the authors wrote "where $F$ can be any class of functions that satisfying the integrability constraints of the theorem."  I think the authors mean "where $F$ can be any function satisfying the integrability conditions of (7)". Indeed, the class of function reduced to the null function satisfies the integrability constraints, but then, equality (7) does not hold.

(ii) Before Eq (8), the authors wrote "by applying the law of large numbers" : I do not think they "applied" the theorem, but get inspired by it to derive the estimator.

(iii) In Proposition 4, one can not take $\alpha=0$.


Notational remarks:
- We do not know what is $T_\theta$ in (8).
- Notations are not consistent. Example, among many others : $\mathbb E$ and $E$ denoting the expectation.  Same remark for $P(X,Y)$ and $P_{X,Y}$.

**Strengths And Weaknesses:**

Strenghts :

1° The authors give a clear introduction on the problem of estimating mutual information.

2° The authors clearly present their numerical experiments so that it is easy to follow.

3° Extensive and quite convincing numerical experiments are provided. Clear guidelines on what kind of scale-invariant estimators one should use are given.

Weaknesses:

1° The second section lacks information about the different estimators. Some terms are not defined, E.g., the joint Shannon entropy is not defined. More importantly, the k-NN distance should be defined, as well as $n_{x,i,\infty}$ which appears in Eq. (6) but is not defined. This section should be self-contained.

2° The proofs of Section 3 are just "proof sketches". It looks like a draft at some points, in the sense that it is very hard to verify the proof precisely, especially since some terms are not defined in the paper (see Weakness 1°).

In particular the proof sketch of Proposition 5 is not convincing for the following reasons :

a) In (8), MINE is defined as $\max_\theta$. In view of (7), I would define MINE as $\sup_\theta$. Why does this maximum exist ?

b) If this maximum does not exist, one can not consider $f^*$ in the proof of Proposition 5.

c) I tried to find the reference (Ghosh et al 2019) to verify the bound $|W_T|^2\le\gamma\alpha^2|T|$, but did not find it (by the way, what is $\gamma$?). Could you state this result? It is difficult to verify this bound, since we do not know what type of neural networks architectures is considered.
Could you also explain with more details how $|W_T|\to 0$ implies $\lim_\alpha\hat I_{MINE-sgd}^n(X;\alpha T)=\hat I_{MINE-sgd}^n(X;0)$ ?

3° The authors often claim that their result are significant, but the reader can not check it easily, since no confidence interval appear on the figures.

---

> ### Author Response · Authors · 2025-01-23
> **Response (Part 1)**
>
> We thank the reviewer for their useful and constructive feedback on our submission. Below, we address the concerns raised, and we have made corresponding revisions in the paper.
>
> **Weakness 1: The second section lacks information about the different estimators. Some terms are not defined, e.g., joint Shannon entropy, k-NN distance, should be defined, as well as $n_{x,u,\inf}$  which appears in Eq. (6) but is not defined. This section should be self-contained.**
>
> We agree with the reviewer that more details are necessary to ensure this section is self-contained. In our revised manuscript:
>
> - We have explicitly defined the joint Shannon entropy in Section 2.
> - We added a detailed explanation of the k-NN distance and the term $n_{x,i,\infty}$ as it appears in Eq. (6).
> - We revised the second section to ensure that all important terms and concepts are properly defined and that the reader can follow the discussion without needing external references.
>
> **Weakness 2: Proof sketches in Section 3 are insufficiently detailed, particularly Proposition 5.**
>
> We acknowledge the reviewer's concern and have substantially expanded the proof sketches in Section 3, redrafting them as proofs, and including them in the appendices.
>
> - For Proposition 5’s proof (which is now Proposition 6), we have now added a separate proposition (Proposition 5) in the revised text that extends the result in Ghosh et al. 2019 specifically to the case of the MINE optimization problem. The proposition 5 in the revised manuscript addresses the limiting case when $\alpha \rightarrow 0$ in the MINE optimization problem. We provide an intuitive argument for why the weights associated with $\alpha T$ in the first layer tend to 0 as $\alpha \rightarrow 0$.  From eq. (12) of the updated manuscript, we see that the gradient descent based updates of the weights are proportional to $\alpha$. The rest of the proof essentially shows that the other terms in (12) are bounded. Thus, when $\alpha \rightarrow 0$, the updates to the weights associated with $\alpha T$ will be negligible, and final trained weights corresponding to $\alpha T$ will effectively be 0. Note that there are few assumptions that are required for Proposition 5 to hold, including the boundedness of inputs and the network weights, and the assumption that the initialized weights are insigificant in magnitude and very close to 0.
>
> - We provide further clarification regarding the point of why $W_T \rightarrow 0$ implies that effectively the network is estimating $I(X;0)$. This holds mainly because when $W_T \rightarrow 0$, the network essentially ignores the changes in $\alpha T$, and thus $\alpha T$ is essentially treated as a constant variable.
>
> - We will change the definition to $\sup_{\theta}$ instead of $\max_{\theta}$. Regarding the existence of the maximum, if there is no unique maximum, there will be multiple $f*$, and for all of them, the corresponding weights $W_T\rightarrow 0$ according to the argument in the proof of Proposition 5. Thus, the result will follow. This has been updated in the revised version.
>
> - For proposition 3, we have added more details to support the claims when $\alpha \rightarrow 0$, specifically the results concerning $\rho_{k,i,p}$ and $\rho’_{k,i,p}$.
>
> These changes should make the proofs more rigorous and easier to verify.
>
> **Weakness 3: No confidence intervals on figures.**
>
> Thank you for pointing this out. We have now included confidence intervals in all the relevant plots in our experimental results section. Thus, all plots in Figures 8 and 9 have been updated with confidence intervals to demonstrate the overall degree of robustness of our results on real datasets. Also, relevant plots for the synthetic experiments have been updated with confidence intervals. An important finding from our confidence intervals is that while KSG maintains its low variance in its estimations, MINE estimates show significant increase in variance when the scaling factor of one of the variables increases, leading to much larger confidence intervals.

---

> > ### Author Response · Authors · 2025-01-23
> > **Response (Part 2)**
> >
> > **Requested Change A: Organizational improvements regarding experimental sections.**
> >
> > We appreciate this suggestion. We have reorganized the experimental sections to improve the flow:
> > - We have currently moved all experimental results into a single section (Section 5), and we have also ensured that all appropriate definitions (such as the base distributions of additive Gaussian and correlated Gaussians) are located before they are used in any experiment.
> > - We have also simplified the large Tables (Table 2 and 3 of the original submission) into four smaller tables (Tables 2, 3, 4, 5), and for greater interpretability, we have only included the normalized root-mean-squared-error results in the main body of the paper, and shifted all Spearman correlation and bias results to the appendix.
> >
> > **Requested Change B: Lack of clarification in definitions on page 7.**
> >
> > We have rewritten the definitions on page 7 for clarity. Specifically:
> > - We have separately defined the problem setting for the three definitions in the beginning of that section for better organization.
> >  - We clarified the notation and addressed the expectations over $S$ to avoid ambiguity regarding the random variable $X$.
> >
> >
> > **Minor remarks and notational remarks**
> >
> > We have addressed all the minor remarks and notational inconsistencies highlighted by the reviewer:
> >  - About Equation (7), We have revised the text to clarify that $F$ must satisfy not only the integrability constraints but also contribute meaningfully to the optimization problem. This ensures that trivial functions, such as the null function, are excluded, addressing your concern. The revised text now reads:
> > "where $F$ is any measurable function from $\mathcal{X} \times \mathcal{Y} \rightarrow \mathbb{R}$ satisfying the necessary integrability constraints for the two expectations in Eq. (7) to be well-defined. The supremum is taken over all such functions $F$ that contribute meaningfully to the optimization problem, ensuring the validity of Eq. (7)." We hope this revision resolves the ambiguity and clarifies the scope of $F$.
> > - We have addressed the language for the law of large numbers appropriately.
> > - We agree with the reviewer, for Proposition 4, we actually didn’t intend to consider $\alpha =0$, as $\alpha \in \mathbb{R}^+$ will not include zero. We will incorporate the definition of $\mathbb{R}^+$ specifically to clarify the set.
> > - We have clarified the missing notations such as $T_\theta$ in (8) and made sure all notations (e.g. the notation for expectation) are consistent.
> >
> > We hope these revisions address the reviewer’s concerns, and we are grateful for the constructive feedback. We believe that the changes have significantly improved the clarity and presentation of the paper.
> > Thank you again for your time and consideration.

---

### Review · Reviewer_Q8K3 · 2024-10-14

**Summary Of Contributions:**

This is an insightful paper. It explores the idea that while mutual information satisfies the data processing inequality, estimates of mutual information from finite samples do not. This work studies one particular data transformation (scaling by a constant) for which existing mutual information estimators behave undesirably. After some theoretical and empirical demonstrations of the failure modes of existing estimators, the paper proposes and characterizes a possible solution by preprocessing/normalizing input data. Finally, the approaches are used to study training dynamics in neural networks.

**Audience:**

Yes

**Broader Impact Concerns:**

It would not hurt to include a broader impacts statement. MI estimators are used in algorithmic fairness research.

**Claims And Evidence:**

No

**Requested Changes:**

I have denoted revisions I see as essential with (*). Note that points 4-6 can be resolved by making code available.

1. (*) Discuss relationship to “equitability” results from Kinney and Atwal, 2014 and perhaps other related work.
2. Discuss relationship to the benchmarking work of Czyz et al., 2023 more carefully.
3. Discuss relationship to max-sliced mutual information and neural network based “preprocessing”
4. (*) Make experimental details (such as number of samples and covariance structure) explicit for all experiments.
5. (*) Present experimental setup in a more centralized way (e.g. in figure captions).
6. (*) Provide detail on implementation of MINE estimator.
7. Condense main text of paper, moving non-essential results and discussion to appendix.
8. Reorganize the paper for clarity.

**Strengths And Weaknesses:**

##

**Strengths:**

- The paper makes a strong point that existing estimators are far from accurately reflecting the data processing inequality. Scale invariance is one of the most basic properties we expect of mutual information, and this paper shows in detail that popular estimators fail to satisfy it.
- The results are practically relevant and useful for implementations of mutual information estimators. It is very common to use what this paper calls “local normalization” and this paper shows that in some scenarios, it can actually degrade the performance of nearest-neighbor estimators.
- The study of training dynamics in Figure 7 is very instructive as a “cautionary tale” showing that  choice of MI estimator has a huge influence on results in practice.

**Weaknesses**:

The paper has several major, **though resolvable**, weaknesses, which can be broadly categorized into (1) contextualization with existing work, (2) lack of clarity about experimental details, (3) presentation issues, (4) miscellaneous.

*Existing work*

- I am surprised that there is no discussion of prior work on the equitability of dependence measures, for example Kinney and Atwal, 2014. Scale invariance is one particular aspect of equitability, and results from the study of equitability are clearly relevant here.
- The introduction of the paper briefly mentions the recent benchmarking work of Czyz et al., 2023, and states that the preprocessing approaches studied by Czyz et al. are equivalent to local normalization. In my opinion, it warrants a more nuanced discussion. For example, in that paper, why do some local normalizations work better than others (e.g. Gaussianization vs uniform marginals)? Does that lend any insight into choice of normalizations here? There may be other more relevant points to focus on, but in general I think the paper deserves more thorough discussion, as it is so closely related.
- The maximization step proposed for KSG is interesting, and is closely related to the idea of max-sliced mutual information (Tsur et al., 2023). Generalizing the idea of maximizing over a scalar $c$, one could imagine maximizing over linear projections. While Tsur et al., 2023 considers projections to a smaller $k<<d$, this paper suggests there is value in max-slicing where $k = d$. I think this connection is worth discussing or at least mentioning as related work.
- Beyond the local/global normalizations proposed in this work, one can imagine many kinds of data preprocessing steps, for example using neural networks to map input data to distributions more amenable for MI estimation. This is an area of recent interest —  see Butakov et al., 2024 and Gowri et al.,  2024. While these are of course much more computationally costly than simple rescaling, it is worth discussing the advantages and disadvantages of both approaches.

*Experimental details*

- In general, I find the details of empirical evaluations to be very unclear. I will give some particular examples below (though not exhaustive). To me, these are critical flaws which cannot be ignored. A simple solution to most if not all the comments below is to make an annotated repository of code which can be run to reproduce the results of the paper. It can be anonymized for peer review in a number of ways. Beyond strengthening the paper, this will also be useful to the community.
- General comment: your readers will appreciate if you put the experiment details in the figure captions. It is inconvenient to have to search for the experimental details in the text of the paper. It is possible that some of the missing details listed below may actually be in the text — but I was unable to find them after considerable time spent searching.
- Figure 1 experiment has many missing details, for example: samples per estimate, evaluated noise dimension numbers (e.g. is it every 5?; currently unclear due to lack of line markers), the values of $\sigma$ and $\sigma’$, number of trials.
- Figure 2: same comments as above. Also, the switch from additive Gaussian in Figure 1 to multivariate Gaussian with random $\rho$ in Figure 2 is confusing to me, though not unsound. It is not clear what is the “certain range” for $\rho$. Ideally, you can provide the readers with the covariance matrix, perhaps in the appendix. Finally, since this is specified as a mean of 10 trials, it would be helpful to give some information about the distribution, e.g. s.d.
- Figure 3:  Similar comments about the choice of $\rho$. As an aside, I do not think this is essential to have in the main text. It is nice to see, but I would personally be satisfied by a reference to Czyz et al, 2023 and a pointer to the appendix.
- You can generalize these comments for Figures 4-7.
- The details for Section 6.3 were hard to find. They are spread across 6.1, 6.3, 6.3.1, 6.3.2, and Tables 2 and 3. It would be easier for the reader if they were more centralized.
- While reading this paper, a natural question to ask is how scale invariance is affected by dimensionality and sample number. Because of the lack of clarity with experimental details, it is  hard, if not impossible, to get any sense for this.
- For all MINE estimates, you must specify what the critic architecture and training procedures were.
- For MINE, are you returning the estimate from train samples or held-out test samples? This is critical to know, especially for the observation of positive bias. If you make the estimate from training samples, you often see a positive bias (which is not there for estimates from held-out test samples). Czyz et al., 2023 carefully reports these kinds of details in the appendix, and it would be worthwhile to take similar care here.

*Presentation issues*

- The paper is quite long, and in my opinion, a lot of it could be moved to the appendix without detracting from the main message of the paper. I strongly encourage the authors to consider condensing the work for clarity.
- Tables 2 and 3 were very tough for me to parse. It would be more pleasant to have a concise summary figure or table in the main text, and the full table in the supplement. Czyz et al., 2023 shows one way of making a large numerical table easy to read.
- I found the structure of the paper quite confusing. The paragraph at the end of the intro describing the structure of the paper is already quite hard to follow, and strangely, omits section 5. I strongly encourage the authors to simplify the organization of the paper. For example, trying to condense and aggregate results into fewer sections.
- It is hard to read the information plane plots in Figure 10. Consider making line plots of the form of Fig. 5 in Butakov et al., 2023.

*Other notes*

- A critical assumption that underlies reasoning about local normalization is that “noise” (which is the unshared information in each variable) is at a smaller scale than the mutually informative structure. It is not immediately clear that this is true in all real world data. It is important that this assumption is carefully discussed. Can we think of situations where it doesn’t apply?
- I do not understand the proof of proposition 5, as it relies heavily on a result from a different paper which I have not read. I would encourage the authors to at least restate the relevant theorem.
- Proposition 3 says that KSG is not scale invariant for very small and very large $\alpha$, but in practice we mostly care about scale invariance in the non-asymptotic regime. It is worth at least mentioning this.

*References*

Kinney, J. B. & Atwal, G. S. Equitability, mutual information, and the maximal information coefficient. Proc. Natl. Acad. Sci. U. S. A. 111, 3354–3359 (2014).

 Czyż, P., Grabowski, F., Vogt, J. E., Beerenwinkel, N. & Marx, A. Beyond normal: On the evaluation of mutual information estimators. arXiv [stat.ML] (2023)

Tsur, D., Goldfeld, Z. & Greenewald, K. Max-Sliced Mutual Information. arXiv [cs.LG] (2023).

Butakov, I., Tolmachev, A., Malanchuk, S., Neopryatnaya, A. & Frolov, A. Mutual information estimation via normalizing flows. arXiv [cs.LG] (2024).

Gowri, G., Lun, X.-K., Klein, A. M. & Yin, P. Approximating mutual information of high-dimensional variables using learned representations. arXiv [q-bio.QM] (2024).

Butakov, I. et al. Information Bottleneck analysis of deep neural networks via lossy compression. arXiv [cs.LG] (2023).

---

> ### Author Response · Authors · 2025-01-23
> **Response (Part 1)**
>
> We would like to thank the reviewer for their detailed comments and constructive feedback. We appreciate the positive remarks regarding our work. Below, we address each of the reviewer's concerns and have made corresponding revisions in the paper to clarify and improve the manuscript.
>
> **Weakness 1: discussion with existing works**
>
> We really appreciate the reviewer's suggestion to expand the discussion on related work, which are quite interesting and relevant. These are our overall observations and thoughts on the suggested works:
>
> - **Relationship to Equitability (Kinney and Atwal, 2014):**
> Thanks for sharing this important work, we have gone through it in detail, and have dedicated a paragraph in the revised version that summarizes this work and other related works on equitability. It seems that this work on equitability is the first that proposes the notion of self-consistent equitability, which is a generalized form of one-sided scale invariance discussed in our work, as we consider the specific case when the function is scale related. In our work, however, we would say that we do not focus on the invariance to data transformation aspect generally, but mainly focus on the specific behaviour of estimators in response to scaling. In their and other related works, they choose a very wide range of functions (Table 2 of [1]) to get a broad overview of the behaviour of MI in response to any type of transformation. However, we noticed that scaling wasn’t one of them, perhaps because they were already locally normalizing the data before sending them to the estimator, which makes the measures already self-equitable with scale.
> Furthermore, self-consistent equitability is only indirectly tested in our experiments in Tables 1 and 2, where we cascade a range of transformations and see the resulting estimation errors for the various compared metrics. Our focus therefore is mainly here on preserving the accuracy of the estimators in response to scaling. Also, we have noted that our choice of transformations (apart from cube and sigmoid) are very different in nature when compared to Table 2 of [1], as they either encompass all dimensions (like invertible random matrix multiplication) or change the dimensionality of the input by adding noisy dimensions or duplicate dimensions. In contrast, every transformation in Table 2 of [1] is applied individually to each dimension of the input. Lastly, as we cascade a random subset of these operations in a random order, we get a richer set of MI-preserving transformations, which can potentially also be tested in an equitability setup using their choice of metrics (like in Figure 2 of [2]). We are considering this for future work.
>
> [1] Reshef, D., Reshef, Y., Mitzenmacher, M., & Sabeti, P. (2013). Equitability analysis of the maximal information coefficient, with comparisons. arXiv preprint arXiv:1301.6314.
>
> [2] Kinney, J. B., & Atwal, G. S. (2014). Equitability, mutual information, and the maximal information coefficient. Proceedings of the National Academy of Sciences, 111(9), 3354-3359.

---

> ### Author Response · Authors · 2025-01-23
> **Response (Part 2)**
>
> - **Further Contextualization with Czyz et al., 2023:** Yes, we concur with the reviewer that it is important to outline a more detailed summary of Czyz et al., 2023, and compare with our work. Below we outline three different aspects which we summarize for each work and compare.
>
> **Choice of transformations:**  As the focus in Czyz et al., 2023 isn’t particularly on the neural network use case where $X$ is the input and $Z$ represents a feature layer, their choice of transformations is motivated differently. Most of their transformations are dimension-wise, i.e. a transformation applied to each dimension. The only exception is their spiral diffeomorphism (Figure 5 of Czyz et al. 2023), which radially morphs the distribution in such a way that the MI is preserved.
> We note that as our focus is mainly on the natural use cases of MI in deep learning, we construct certain types of transformations relevant to this setting. Apart from the cubic transformation, which is dimension-wise, all our other transformations are motivated by the potential use case in deep learning. We outline each one as follows. The sigmoid transformation is motivated by potential uses of sigmoid in the network hidden layers. The random matrix multiplication (randmat) transformation is motivated via the features undergoing similar transformations through neural network layers, which are usually matrix multiplications followed by a non-linearity. Note that the randmat also has an additional scaling term $\alpha$ (Section 6.3 of the paper), which scales the resulting transformed vector as we wish to also focus on the robustness to scale. The duplicate-noise transformation, which adds dummy noise dimensions, is motivated via the fact that the number of hidden neurons can change through the layers and often many of these dimensions deeper within the network are usually very sparse and noisy. Similarly, the duplicate-self transformation is another approach to changing the dimensionality of the input while preserving the total information.
>
> **Preprocessing methods:** The authors indeed find that the Gaussianization based local normalization yields better results than uniform marginalization, when the base distribution is the multivariate student distribution. However, they do note that the improvements are minor, and overall they end up choosing the standard variance based local normalization over the other two. Our conjecture is that Gaussianization may work better when the distribution has long tails, as long tail distributions are typically harder to estimate MI for, which was observed in Czyz et al., 2023. This is potentially because the data samples which are a part of the long tail end of the distribution may end up adding more noise to the final estimate, than a typical case with Gaussian variables, and Gaussianizing the data preserves MI while avoiding long tails in the input, thereby leading to better performance. We agree with the reviewer that similar considerations can be taken in our global normalization strategies to further enhance the accuracy of MI estimators, which is something we are considering for future work.
>
> **Scale invariance:**  As the data was already preprocessed using local normalization approaches, scale invariance isn’t a part of the analysis in Czyz et al., 2023, similar to Kinney and Atwal 2014. In our case, all our normalization approaches are based on ensuring that the MI estimates are scale-invariant and while retaining other desired properties which may not hold for standard normalization approaches, like robustness to noisy dimensions.
>
> - **Max-Sliced Mutual Information (Tsur et al., 2023):** We agree that max-sliced mutual information also has the concept of maximizing MI across the projections. However, our observation is that the ground truth MI values across the slices are usually different, and the max-sliced MI is naturally upper bounded by the true MI. Furthermore, the projections are of unit norm, and thus they don’t scale the data to different degrees of energy. In our case, our maximization is mainly motivated by the aspect that the ground truth MI stays the same in response to scaling, whereas the finite sample MI does not, as it drops to negligible values when the scale of the data is at either extremity. We found that our approach is similar to the use of maximization in KDE based MI estimation in [3], where the KDE bandwidth was chosen such that it maximized the MI estimate between two random variables. It seems that this idea of maximizing dependence has been used in statistical tests to detect independence of two random variables (Horowitz et al. 2001, Heller et al. 2016).
>
> [3] Zeng, X., Xia, Y., & Tong, H. (2018). Jackknife approach to the estimation of mutual information. Proceedings of the National Academy of Sciences, 115(40), 9956-9961.

---

> ### Author Response · Authors · 2025-01-23
> **Response (Part 3)**
>
> - **Neural network-based preprocessing (Butakov et al., 2024; Gowri et al., 2024):**
>
> These are interesting recent papers that approach the problem of MI estimation differently. We have gone through them and have included discussions on them in the revised version. These are the summary of our observations from these works.
>
> **Gowri et al. [4]:** In this work, the authors propose a two-step MI estimation procedure. First, they train compressed representations of the input which minimize an upper bound on the conditional entropies between the compressed representations and the input/output, after which they estimate the MI between the compressed representations using KSG.  In the limiting case, these compressed representations would preserve the true MI, otherwise normally they are upper bounded by it. It is worth noting their compressed representations can be of any isomorphic form w.r.t the choice of compressor, and as they use neural networks, the scale of the compressed variables can be arbitrary. So to alleviate the issue, they use the standard unit normalization approach (local).  As such, our findings in this work, including the different variants proposed for global normalization, can be applied to their KSG-based MI estimation using the compressed representation to generate more robust MI estimates.
>
> **Butakov et al. [5]:** It seemed to us that this work proposed a new estimator using normalizing flows, which in this case are function maps (which can be a neural network) which eventually transform the data distribution to a simpler one which has a closed form solution for MI estimation. In their work, they seem to focus on a Gaussian base distribution. In principle, their estimator could be scale invariant, as their final estimator is the analytical MI expression itself. However, we did not see any explicit theoretical study on this, as one will need to show that the correlation matrix at the end of applying multiple normalizing flows is invariant to one-sided scaling of one of the variables. Overall, our work points out that scale-invariance is an important consideration for any MI estimator to prevent scale confounding in the estimates.
>
> [4] Gowri, G., Lun, X. K., Klein, A. M., & Yin, P. (2024). Approximating mutual information of
>                     high-dimensional variables using learned representations. arXiv preprint
>                     arXiv:2409.02732.
>
> [5] Butakov, I., Tolmachev, A., Malanchuk, S., Neopryatnaya, A., & Frolov, A. (2024). Mutual
>                     information estimation via normalizing flows. arXiv preprint arXiv:2403.02187.

---

> ### Author Response · Authors · 2025-01-23
> **Response (Part 4)**
>
> **Weakness 2: Clarity of Experimental Details**
> We understand the importance of clear and accessible experimental details, and we thank the reviewer for the specific recommendations provided. We outlines the we have made the following changes:
> - **General Comments - code submission:**
> We have attached our codes required to reproduce all of our experiments in supplementary material. The zip file includes annotations and instructions to reproduce our empirical results. We believe this will address many concerns about reproducibility.
> - **Detailed Descriptions in Figure Captions and In-text Experimental Detailed Descriptions:** Due to space constraints on the inline Figures, we are unable to insert all experimental details in the captions. However, currently, we have added all relevant details for each Figure in a centralized way in Appendix A, and we refer to it within the captions. We are including all relevant details such as sample sizes, noise parameters ($\sigma$, $\sigma’$), ranges of evaluated noise dimensions, covariance matrix, the parameter $\rho$ and its range when applicable, number of trials, and distributions of results (e.g., standard deviation across 。trials).
> - **Switching between additive Gaussian noise and correlated Gaussian noise:**
> In our experiments, we consider these two base distributions, and for our experiments in section 5, we choose of mixture where some of the settings are additive Gaussian noise based, and the others are correlated Gaussian noise based. We do this to better showcase the differences between the different normalization variants, as we find that for certain distribution bases, the differences are more apparent for MINE (correlated Gaussian), whereas for KSG, additive Gaussian noise base yields more pronounced trends.
> - **Improving the Organization of Section 6.3 (Now Section 5.2) for Clarity:**
> We appreciate the reviewer’s feedback on the organization of Section 6.3. To improve clarity and accessibility, we have consolidated the relevant information by restructuring the section. We now present all key details in a single, unified subsection, minimizing cross-references. Additionally, we have clarified the placement of Tables 2 and 3 within the section for easier navigation.
> - **Clarifying Scale Invariance in Relation to Dimensionality and Sample Size:**
> To address how scale invariance is influenced by dimensionality and sample size, we have added a new experiment specifically designed to analyze how the scale invariance analysis plots in Figure 6 of our paper change in response to changes of dimensionality and sample size. Currently for figure 6, we use a sample size of 1000 (similar scale as our experiments in Table 1 and 2), and a dimensionality of 2. In our new experiments, we have added more sample sizes $N=10000,50000$ and dimensionality $d=10,50$. Currently, the KSG results have been added only, but we will also incorporate the MINE results for the final version.
> - **Specifying Critic Architecture, Training Procedures, Use of Train vs. Test Samples for MINE Estimates:** In response to the reviewer's request, we have specified the details of the MINE estimator implementation, including the actual architecture, training parameters in the revised manuscript (Appendix B.2). Additionally, our shared code contains the files used to run all experiments, and includes the MINE libraries and all details. These clarifications aim to address questions regarding the bias of our MINE estimators and to align our setup with the practices demonstrated in Czyż et al. (2023). We do not estimate MINE on a separated validation set because we mainly focus on the small sample size regime, which reduces the training set further and gives a very small validation set. Furthermore, we do see in our experiments in Tables 2 and 3 that our MINE estimators, similar to KSG show an increasingly negative bias with greater dimensionality, and thus avoid over-estimation. Thus, we employ the conventional approach of computing the MINE estimate as the training data loss.

---

> > ### Author Response · Authors · 2025-01-23
> > **Response (Part 5)**
> >
> > **Weakness 3: Presentation and Structural Improvements**
> >
> > We appreciate the reviewer’s feedback on improving the presentation and organization of our paper. We have made the following changes to enhance clarity and accessibility:
> >
> > - **Condensed Main Text and improving readability of Table 2 and 3:** We have streamlined the main text by moving non-essential results and more detailed discussions to the appendix, allowing us to focus on the key findings in the main sections.  The proofs sketches have been made more detailed into proper proofs and moved to the Appendix. The results in Tables 2 and 3 have been compressed and only the normalized rmse of measures for the Additive Gaussian distribution base are now reported in the main paper, to make the main results more readable and interpretable. We have also separated the results for the KSG-based estimators and the MINE-based estimators and created two separate tables. The full results are now moved to the Appendix.  The information plane results and discussions have been moved to the Appendix. Note that the resulting paper still has 18+ main pages of text as we have added significant discussions with other related works based on your suggestions, and elaborated on some definitions which were unclear to other reviewers.
> >
> > - **Reorganized Structure for Enhanced Clarity:** We have revised the structure of the paper to improve readability and logical flow. This includes explicitly adding Section 5 to the roadmap paragraph in the introduction, ensuring consistency and coherence throughout the paper. Related sections have been combined where possible to create a streamlined structure.
> >
> > - **Enhanced Readability of Information Plane Plots:** To address the readability of our information plane plots, we have revised these plots in the style of line plots (Figure 11 in Appendix). This change makes these visualizations more accessible and clarifies key findings in training dynamics.

---

> ### Author Response · Authors · 2025-01-23
> **Response (Part 6)**
>
> **Other notes:**
>
> - **Discussing the Assumption on Noise Scale in Local Normalization**
> Yes, the assumption that the scale of noise dimensions is smaller than the scale of the informative dimensions of the input is present in our experiments that motivate the use of global normalization over local normalization (e.g. Figures 1 and 2). We do specify this assumption in the text with a short reasoning which states that for neural networks irrelevant hidden neurons typically should have less energy than the relevant ones.  Our overall assumption is that given two RVs $X$ and $T$, we assume that the very low energy dimensions of $X$ contribute minimally to the mutual information $I(X;T)$. In the general case of any two continuous RVs $X$ and $T$, it is definitely possible that low-energy dimensions of $T$ will relate to $X$ and can in principle form the bulk of the mutual information $I(X;T)$. However, in the neural network case, low energy feature dimensions in $X$ typically will constitute a very negligible part of the mutual information $I(X;T)$, as neural network functions are typically Lipschitz, and small changes in $X$ will also elicit a very small change in $T$. Similarly, smaller dimensions of $X$ will also elicit a smaller gradient, as the gradient descent updates are proportional to the magnitude of the input. This also mirrors the overall MINE result, which finds that as the scaling factor $\alpha\rightarrow 0$, the weights associated with $T$ in the estimation of $I(X;\alpha T)$ are ultimately very close to zero (Proposition 5 of revised paper), and thus $T$ ends up minimally affecting the output of the network. As $X$ and $T$ are functionally related, it is more likely that very low energy dimensions of either contribute minimally to $I(X;T)$.
>
> - **Restating the Theorem for Proposition 5 for Clarity**
> For Proposition 5’s proof (which is now Proposition 6), we have now added a separate proposition (Proposition 5) in the revised text that extends the result in Ghosh et al. 2019 specifically to the case of the MINE optimization problem. The proposition 5 in the revised manuscript addresses the limiting case when $\alpha \rightarrow 0$ in the MINE optimization problem. We provide an intuitive argument for why the weights associated with $\alpha T$ in the first layer tend to 0 as $\alpha \rightarrow 0$.  From eq. (12) of the updated manuscript, we see that the gradient descent based updates of the weights are proportional to $\alpha$. The rest of the proof essentially shows that the other terms in (12) are bounded. Thus, when $\alpha \rightarrow 0$, the updates to the weights associated with $\alpha T$ will be negligible, and final trained weights corresponding to $\alpha T$ will effectively be 0. Note that there are few assumptions that are required for Proposition 5 to hold, including the boundedness of inputs and the network weights, and the assumption that the initialized weights are insigificant in magnitude and very close to 0.
>
> - **Noting Practical Implications of KSG's Scale Invariance in Proposition 3 (Proposition 3 says that KSG is not scale invariant for very small and very large α, but in practice we mostly care about scale invariance in the non-asymptotic regime. It is worth at least mentioning this.)** Yes, we have added this point in the text now. Given that, we found it interesting that in our scale invariance experiments, the average MI estimates showcase a significant drop with small changes in scale (less than 10x) (Figure 6).
>
> **Broader Impact Statement**
> Following the reviewer's recommendation, we have added a Broader Impact Statement to discuss potential societal implications of MI estimators, particularly within applications like algorithmic fairness.
>
> Once again, we thank the reviewer for their detailed feedback, which has been invaluable in strengthening the clarity of our paper. We look forward to implementing these improvements and believe they will enhance the accessibility and impact of our work.

---

### Review · Reviewer_SDtz · 2025-01-10

**Summary Of Contributions:**

This paper emphasizes the critical role of scaling in mutual information (MI) estimation, highlighting a fundamental limitation in widely-used methods. MI, by definition, is invariant to transformations like scaling, but many estimators fail to preserve this property.

To address these shortcomings, the paper introduces global normalization strategies, which scale the variables as a whole while preserving the relative energies between dimensions. By applying these strategies, the new KSG-Global-L∞ and MINE-Global-Corrected estimators achieve true scale invariance. These approaches ensure that MI estimates remain consistent across a wide range of scaling factors, improving reliability in scenarios where variable scales change due to preprocessing or data transformations.

**Audience:**

Yes

**Broader Impact Concerns:**

No concerns

**Claims And Evidence:**

No

**Requested Changes:**

1. Clarify the role of \alpha = 0: Provide a more detailed explanation of why mutual information is considered well-defined when \alpha = 0. Address the apparent contradiction in Proposition 5, where the MINE-SGD estimator’s failure to maintain scale invariance is tied to MI tending to zero. This requires further theoretical justification or experimental support.
2. Discuss computational overhead
3. Improve real dataset results presentation

**Strengths And Weaknesses:**

### **Strengths**
1. The paper addresses a significant yet underexplored issue in mutual information (MI) estimation: scale sensitivity.
2. The discussion of the pitfalls of local normalization highlights the limitations of current practices and motivates the need for global normalization.
3. The introduction of KSG-Global-L∞ and MINE-Global-Corrected demonstrates originality, offering effective ways to achieve scale invariance while improving MI estimation accuracy.
4. The authors conduct thorough experiments on synthetic and real-world data, validating their methods across various settings.

---

### **Weaknesses**
1. While the theory addresses all scaling factors (\alpha), including \alpha = 0, this raises questions about whether mutual information is well-defined in this extreme case. Proposition 5 indicates that the MINE-SGD estimator is not scale-invariant because the MI estimate tends to zero as \alpha \to 0. This argument is not fully convincing, as it is unclear why MI itself would not also tend to zero in this scenario.
2. The additional steps of the new estimators, such as maximization in KSG-Global-L∞, may introduce computational costs, especially in high-dimensional data or large-scale datasets. This should be more discussed in the paper.
3. Results on real datasets are not particularly compelling. For instance, in Figure 9, the claim that MINE is noisier than other estimators is not evident from the figure. Similarly, Figure 19, which is meant to showcase the information bottleneck phenomenon, is difficult to interpret due to its small size and overlapping points.
4. The length of the paper, at 19 pages excluding references and appendix, feels excessive for the content presented. While the subject is important, some sections could be streamlined to improve readability and reduce redundancy, ensuring that the core contributions remain the focus.

---

> ### Author Response · Authors · 2025-01-23
> **Response**
>
> We thank the reviewer for their useful and constructive feedback on our submission. Below, we address the concerns and weaknesses raised, and we have made corresponding revisions in the paper.
>
> >While the theory addresses all scaling factors (\alpha), including \alpha = 0, this raises questions about whether mutual information is well-defined in this extreme case. Proposition 5 indicates that the MINE-SGD estimator is not scale-invariant because the MI estimate tends to zero as \alpha \to 0. This argument is not fully convincing, as it is unclear why MI itself would not also tend to zero in this scenario.
>
> Yes, the reviewer has brought to attention an important distinction between the case when $\alpha=0$ when MI is actually zero, and when $\alpha \rightarrow 0^{+}$, when MI still follows one-sided scale invariance rule i.e., $I(X;Y) =I(\alpha X; Y)$. We will note in the writing that our results are not targeted to the former case, but in the limiting case when $\alpha$ is arbitrarily close to 0 but not exactly 0. An additional observation to provide further intuition why as $\alpha \rightarrow 0^{+}$ MI will still follow the scale invariant property, is as follows. As MI is double-sided scale invariant, estimating $I(\alpha X;Y)$ is the same as estimating $I(X;Y/\alpha)$, and as $\alpha$ gets arbitrarily close to zero, this is equivalent to the case when one of the variables is scaled to extremely large magnitudes ($Y$). In this scenario, none of the RVs actually approach zero when $\alpha \rightarrow 0^{+}$, but rather approach very high values towards infinity.
>
> This argument applies to all our theoretical results which study the limiting case of $\alpha$, both for MINE and KSG. Our empirical results reflect these findings, as we find that both for KSG and MINE in the extreme $\alpha$ cases they drop down to very small values, and the drop is not sharp but relatively smooth.
>
> >The additional steps of the new estimators, such as maximization in KSG-Global-L∞, may introduce computational costs, especially in high-dimensional data or large-scale datasets. This should be more discussed in the paper.
>
> Yes, the maximization based approaches with KSG increases computational cost, and we will reflect this in our discussion.
>
> >Results on real datasets are not particularly compelling. For instance, in Figure 9, the claim that MINE is noisier than other estimators is not evident from the figure. Similarly, Figure 19, which is meant to showcase the information bottleneck phenomenon, is difficult to interpret due to its small size and overlapping points.
>
> After perusing some of the comments and suggestions from other reviewers, we have made our results on real-datasets more presentable by adding confidence intervals for each MI estimator estimate at each epoch of training. This stabilizes our results and observations in Figure 9, as we can comment on the general trends with more certainty. For the information plane plots Figure 10 (we assume you meant Figure 10, not 19), we have made some changes to its presentation that we hope improves its interpretability (Figure 11 in revised version). This includes averaging over more trials for more smoother trends and adding lines between consecutive observations to view the trajectory of the information measures better.
>
> >The length of the paper, at 19 pages excluding references and appendix, feels excessive for the content presented. While the subject is important, some sections could be streamlined to improve readability and reduce redundancy, ensuring that the core contributions remain the focus.
>
> Yes, this comment has been reflected by other reviewers as well, and we have currently moved some of the different parts of our work to the appendix. For instance, we have removed the non-rmse results in Tables 2 and 3 and moved those details to the appendix, and we have moved the theoretical proofs to the appendices as well. We have also shfited the information plane diagrams and their discussion to the appendix, as we feel Figure 9 already largely informs us regarding the behaviour of MI in real datasets. However, after incorporating some of the essential discussions with other relevant work and other clarifications on various parts of the paper, the paper length overall is similar as before.

---

### Author Response · Authors · 2025-01-24
**General Response**

We thank all reviewers for their detailed comments and useful suggestions. We summarize the changes made to our submission as follows:

1.   Discussions with other related work have been added, notably equitability, the work by Czyz et al. (Neurips 2023) and other neural network based pre-processing for MI, and all our contributions have been discussed in the context of these prior works.
2. The proof sketches have been made clearer and have been converted into proofs. Instead of the main paper, they have now been shifted to Appendix F.
3. The proof of Proposition 5 (Proposition 6 in revised version), which earlier relied on a result from another paper (Ghosh et al. 2019), now has been replaced with a standalone proof, which uses a new theoretical result (Proposition 5) targeted specifically for the case of the MINE optimization.
3. The paper has been overall re-structured for better clarity and readability. The large, dense tables have been moved to the Appendix and concise versions of them have been incorporated in the main text by separating the KSG and MINE results and only discussing normalized RMSE measures. The experimental sections have been re-organized such that all necessary definitions are outlined prior to discussing any results.
4. The information plane plots have been made clearer by averaging across more trials for smoother plots on all datasets, and the trend is now better visible (Fig. 11)
5.    Empirical details for every experiment has been clarified separately in Appendix A.1, including the configurations used for the KSG and the MINE estimators for each case, and the code for all experiments have been released.
6.   Confidence intervals have been added to the relevant plots, including the results on MI estimation on real datasets.
7. Additional discussions have been added for the results on the synthetic data, regarding some interesting observations on the nature of Spearman correlation and bias of the estimators (Appendix C).
8.   Additional results exploring the effect of one-sided scaling on estimators have been added for different sample number and data dimensionality in the Appendix.
9. All missing definitions in the background section have been added, and the definitions of the estimators have been clarified further. The definitions of the normalization strategies has been made more precise with better organization.
10. All other minor clarifications requested by the reviewers has been incorporated in the text.

---

### Decision · Action_Editor_thoL · 2025-03-09

**Recommendation:** Reject

**Comment:**

While some reviewers are positive about the changes in the manuscript, we have not yet reached a consensus and one reviewers recommends another round of reviews. Since TMLR does not allow major revisions, my recommendation is to reject and I would like to encourage the authors to submit a revised version of their work.

**Audience:**

This will be of interest to the TMLR crowd.

**Claims And Evidence:**

Reviewer SDtz points out that the paper falls short on this -- see their review.

**Resubmission Of Major Revision:**

The authors may consider submitting a major revision at a later time.